# AbstRaL: Augmenting LLMs' Reasoning by Reinforcing Abstract Thinking

**Silin Gao**[1*,2], **Antoine Bosselut**[2], **Samy Bengio**[1,2], **Emmanuel Abbe**[1,2]
[1]Apple  [2]EPFL

## Abstract

Recent studies have shown that large language models (LLMs), especially smaller ones, often lack robustness in grade school math (GSM) reasoning. In particular, they tend to experience performance drops when faced with distribution shifts, such as changes to numerical or nominal variables, or insertions of distracting clauses. A possible strategy to address this involves generating synthetic data to further "instantiate" reasoning problems on potential variations. In this work, we instead focus on the strategy of "abstracting" reasoning problems. This not only helps counteract distribution shifts but also facilitates the connection to symbolic tools for deriving solutions. Focusing on GSM, we find that this abstraction process is better acquired through reinforcement learning (RL) than just supervised fine-tuning, which often fails to produce faithful abstractions. Our method, AbstRaL—which promotes abstract reasoning in LLMs using RL on granular abstraction data—significantly mitigates performance degradation on recent GSM perturbation benchmarks. Besides, improving GSM robustness via AbstRaL is shown to also implicitly benefit LLMs' capabilities on OOD mathematical and general reasoning tasks, indicating that abstract thinking broadly enables better generalizability.

## 1 Introduction

The ability of reasoning, which involves the integration of knowledge to derive dynamic conclusions rather than direct recourse to memorized information (e.g., Yu et al., 2024; Abbe et al., 2024), is an essential quality for artificial general intelligence (Zhong et al., 2024a). Toward this end, recently developed large language models (LLMs) have been equipped with impressive reasoning capabilities, either scoped for general purpose (Guo et al., 2025; Yang et al., 2024a; Grattafiori et al., 2024) or in specialized domains such as mathematics (Shao et al., 2024; Yang et al., 2024b).

However, most LLMs, especially smaller ones[1], still face the challenge of robustness when reasoning, revealing considerable room for improvement in out-of-distribution (OOD) generalization. In particular, recent works (Mirzadeh et al., 2025; Li et al., 2024) have shown that even in simple GSM tasks, LLMs suffer performance degradation when facing perturbations and distribution shifts.

LLMs can be prone to reasoning errors on instantiation shifts such as when numerical or nominal variables of a GSM question are altered, even though the LLMs can respond to the original question correctly. On more challenging interferential shifts, where a distracting (topic-related but useless) condition is added, LLMs suffer even more drastic performance drops.

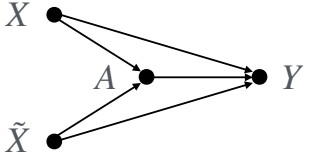

Figure 1: Two paraphrased queries $X$ and $\tilde{X}$, having same solution $Y$, can be both handled by a common abstraction $A$.

To improve the robustness of reasoning, a possible learning strategy (Boix-Adserà et al., 2024) is to synthesize more instances of a reasoning problem that are varied in surface-form contexts but follow the same high-level reasoning schema represented by an abstraction of the problem. In this paper, instead of scaling up the training instances (can be computationally expensive), we teach LLMs to **directly learn an abstraction underlying reasoning problems and thereby learn to reason in a manner**

---

*Work done during internship at Apple.

[1]as verified by our analysis in §5.1

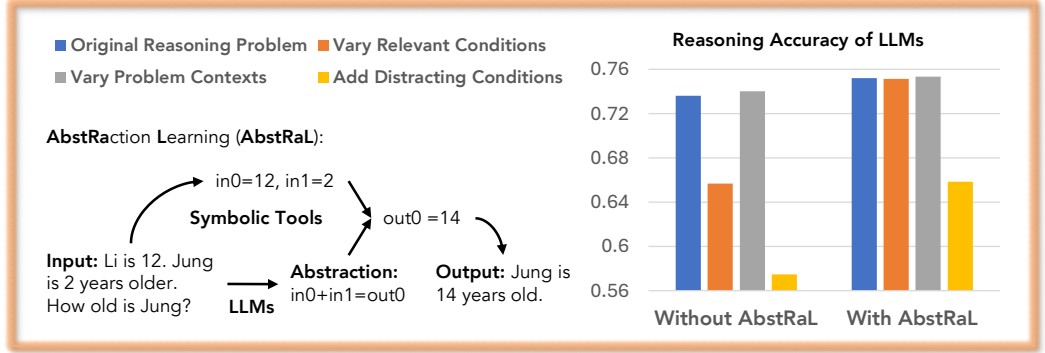

Figure 2: Our **AbstRa**ction **L**earning (**AbstRaL**) method effectively improves GSM reasoning robustness of LLMs, especially facing the variations of relevant input conditions and the interference of distracting conditions. We present average accuracy of all our tested LLMs on GSM-Plus (Li et al., 2024), including the original GSM8K testing set (Original Reasoning Problem), the testing sets with numerical variations (Vary Relevant Conditions), averaged across three portions (digit expansion, integer-decimal-fraction conversion and numerical substitution), the testing set with problem rephrasing (Vary Problem Contexts) and with distractor insertion (Add Distracting Conditions).

**that is invariant to contextual distribution shifts.** Our work is rooted in the abstract reasoning literature (Hong et al., 2024; Dutta et al., 2024; Gao et al., 2025), with a new granular reinforced abstraction learning scheme that promotes the robustness and generalizability aspects of reasoning.

We propose the reinforced **AbstRa**ction **L**earning framework, **AbstRaL**, as shown in Figure 2, which first teaches LLMs to generate mathematical abstraction of input GSM problem, and then connects the abstraction with symbolic tools to stably derive the output solution, regardless of the specific input contexts. The learning of abstraction relies on our **Granul**arly-decomposed **A**bstract **R**easoning (**GranulAR**) data (§3.1) distilled from an oracle LLM, which integrates symbolic reasoning within socratic problem decomposition (Shridhar et al., 2023) and chain-of-thought (CoT) explanations (Wei et al., 2022). On top of supervised fine-tuning (SFT), AbstRaL uses reinforcement learning (RL) with a new set of model-free rewards (§3.2) to further improve the faithfulness of generated abstraction.

We tested AbstRaL on two benchmarks that evaluate the robustness of GSM reasoning (§5.1), GSM-Symbolic (Mirzadeh et al., 2025) and GSM-Plus (Li et al., 2024). Experimental results on various seed LLMs consistently demonstrate that AbstRaL effectively improves the GSM reasoning robustness of LLMs. As shown in Figure 2, AbstRaL almost reverts the performance drop of LLMs caused by variations of relevant input conditions, and also significantly mitigates the interference of distracting conditions added to the perturbed testing samples. Furthermore, we find that LLMs trained with AbstRaL demonstrate improved zero-shot performances on a wide range of OOD tasks within and beyond the math domain (§5.2), such as AIME (MAA, 2024) and BBH (Suzgun et al., 2023). This indicates that, more interestingly, AbstRaL's learning of abstract thinking in the GSM domain can implicitly benefit more general reasoning capabilities of LLMs.

## 2 LEARNING STRATEGIES TO IMPROVE REASONING ROBUSTNESS

We assume that every reasoning data sample, consisting of input question (or query) $\mathcal{X}$ and output answer (or response) $\mathcal{Y}$, is an instantiation of an underlying symbolic abstraction $\mathcal{A}$ that represents high-level reasoning schema. For example, in Figure 3, the problem of calculating Jung's age, according to how much older he is than Li, is based on the abstract arithmetic rule of adding two numbers. A robust reasoner is supposed to master the abstraction $\mathcal{A}$, and therefore stably give a faithful answer $\mathcal{Y}$ to any question $\mathcal{X}$ implicitly derived from $\mathcal{A}$, rather than overfitting to only a subset of instances of $\mathcal{A}$ and vulnerable to distribution shifts that go beyond the subset.

A common strategy to improve the reasoning robustness of LLMs is to augment the learning data by synthesizing more instances $\{(\mathcal{X}', \mathcal{Y}'), (\mathcal{X}'', \mathcal{Y}''), ...\}$ of abstraction $\mathcal{A}$, with paraphrasing (Gan & Ng, 2019; Zhou et al., 2024) or templates (Boix-Adserà et al., 2024). For example, as shown in

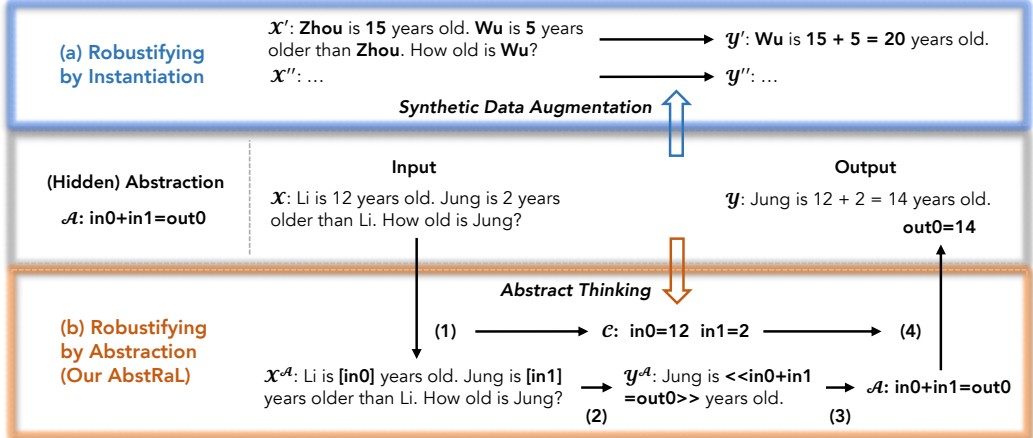

Figure 3: Learning strategies to improve reasoning robustness with respect to distribution shifts. **(a)** Augmenting the amount of learning data by synthesizing more reasoning instances. **(b)** Directly learning to construct the underlying abstraction based on the input, including: **(1)** condition recognition, **(2)** abstract reasoning, **(3)** abstraction retrieval and **(4)** symbolic derivation.

Figure 3 (a), the names and numbers appearing in an instance can be replaced with other values to create a new instance, which has a different problem context and input conditions, but follows the same abstract arithmetic rule. As verified by previous study (Boix-Adserà et al., 2024), this learning strategy requires a large amount of synthetic data augmentation, to effectively boost LLMs' grasp of the high-level abstraction, thus being able to resist the interference of surface-form variations.

In this work, instead, we focus on the strategy of abstract thinking (Hong et al., 2024; Gao et al., 2025), which teaches LLMs to directly learn the abstraction $\mathcal{A}$ underlying each instance $(\mathcal{X}, \mathcal{Y})$, and connect $\mathcal{A}$ with symbolic tools such as an equation solver, to steadily derive the answer $\mathcal{Y}$ to the question $\mathcal{X}$, as illustrated in Figure 3 (b). Without scaling up the training data, our direct learning of abstraction still effectively improves the robustness of GSM reasoning, due to the modeling of more general reasoning patterns represented by the abstraction. We introduce our method in detail in §3.

## 3  ABSTRAL: REINFORCED ABSTRACTION LEARNING FRAMEWORK

Figure 3 (b) illustrates our AbstRaL framework that incentivizes LLMs' abstract thinking in GSM problem solving, which consists of four steps.

**(1) Condition Recognition**    We first parse the input GSM question $\mathcal{X}$ to identify conditions $\mathcal{C}$ that can be used to answer the question, and formulate $\mathcal{C}$ with abstract symbols, which represent the **input variables** in abstraction $\mathcal{A}$. For example, in the problem shown in Figure 3 (b), the numbers 12 and 2 given in the question are used to derive the answer, which are assigned to the symbols in0 and in1, respectively. Then the abstract input question, denoted as $\mathcal{X}^{\mathcal{A}}$, is constructed, by replacing the specific texts (or values) in $\mathcal{X}$ with their assigned abstract symbols in $\mathcal{C}$, enclosed in square brackets. This step can be performed by either a symbolic tool, *e.g.*, a regex-matching script that searches numerical values, or a neural parser designed by prompting an LLM.[2]

**(2) Abstract Reasoning**    We then use our constructed GranulAR data (§3.1) to teach LLMs abstract reasoning on GSM problems, based on SFT and RL (§3.2). In particular, LLMs are tasked to generate an abstract answer $\mathcal{Y}^{\mathcal{A}}$ to the abstract question $\mathcal{X}^{\mathcal{A}}$. In $\mathcal{Y}^{\mathcal{A}}$, LLMs quote input variables in $\mathcal{X}^{\mathcal{A}}$ by their assigned abstract symbols, and also use abstract symbols to represent derived output variables, such as the math derivation result out0 in Figure 3 (b), enclosed in double angle brackets.

---

[2]We introduce the specific experimental implementations of our proposed framework in §4.

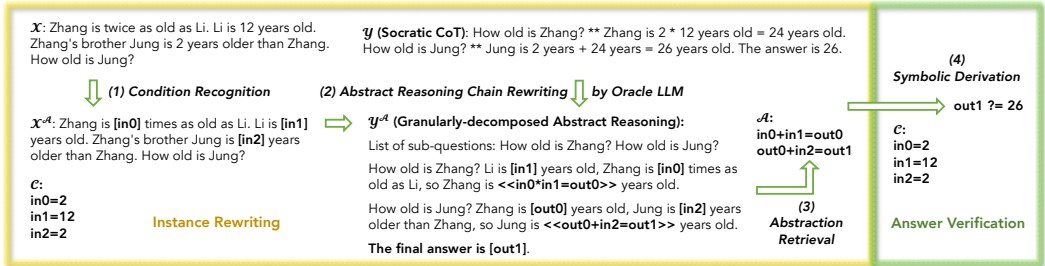

Figure 4: Overview of GranulAR training data construction, which consists of an instance rewriting procedure to rewrite existing socratic CoT data $(\mathcal{X}, \mathcal{Y})$ into fine-grained abstract reasoning data $(\mathcal{X}^{\mathcal{A}}, \mathcal{C}, \mathcal{Y}^{\mathcal{A}}, \mathcal{A})$, followed by a answer verification procedure to check the correctness of rewriting.

**(3) Abstraction Retrieval**    Based on the abstract answer $\mathcal{Y}^{\mathcal{A}}$, we then retrieve the abstraction $\mathcal{A}$ that is de-contextualized from the instance $(\mathcal{X}, \mathcal{Y})$. Similar to the condition recognition, the retrieval of $\mathcal{A}$ can also be done with a regex-matching tool or a LLM with specific prompts.

**(4) Symbolic Derivation**    We finally use the abstraction $\mathcal{A}$, along with the input conditions $\mathcal{C}$, to derive the real answer to the question. This derivation can be accomplished by either a rule-based symbolic parser, *e.g.*, a equation solver for arithmetic, or a neural (LLM) symbolic reasoner.

Our framework decomposes the task of inferring $\mathcal{A}$ from $\mathcal{X}$ into a fine-grained pipeline $\mathcal{X} \rightarrow \mathcal{X}^{\mathcal{A}} \rightarrow \mathcal{Y}^{\mathcal{A}} \rightarrow \mathcal{A}$, which facilitates the learning of this task. Nevertheless, the core abstract reasoning step of inferring $\mathcal{Y}^{\mathcal{A}}$ from $\mathcal{X}^{\mathcal{A}}$ is still non-trivial, which requires fine-grained abstract reasoning data and proper reinforcement learning to achieve robust inferences, as described below.

### 3.1    GRANULARLY-DECOMPOSED ABSTRACT REASONING (GRANULAR) DATA

**Motivations**    LLMs have learned various fine-grained reasoning strategies at either the pre-training (Yang et al., 2024c; Chen et al., 2025b) or the post-training (Kumar et al., 2025) phase, such as chain-of-thought (CoT) (Wei et al., 2022) and socratic problem decomposition (Shridhar et al., 2023) as representatives. In our GranulAR training data, we integrate abstract reasoning with these pre-learned strategies, which enables LLMs to gradually construct the abstraction in a fine-grained reasoning chain, as shown in Figure 4 $\mathcal{Y}^{\mathcal{A}}$. Such data format is close to the pre-training or the post-training data distribution, and therefore eases the difficulty of equipping LLMs with abstract thinking.

**GranulAR Format**    The answer $\mathcal{Y}^{\mathcal{A}}$ of GranulAR first decomposes the question $\mathcal{X}^{\mathcal{A}}$ into a list of sub-questions, which enables a holistic planning of step-by-step reasoning. Based on that, $\mathcal{Y}^{\mathcal{A}}$ answers each sub-question with CoT and abstract symbols, where it first quotes relevant input conditions (or answers to previous sub-questions), and then derives the answer with quoted symbols. Finally, $\mathcal{Y}^{\mathcal{A}}$ draws a conclusion to clarify which output abstract symbol represents the final answer.

**Data Construction**    Figure 4 illustrates how we construct the GranulAR training data. We first conduct the condition recognition described in §3 to formulate the conditions $\mathcal{C}$ from the question $\mathcal{X}$ and creates the abstract question $\mathcal{X}^{\mathcal{A}}$. Based on that, we prompt an oracle LLM to rewrite the gold socratic CoT answer $\mathcal{Y}$ into our desired abstract answer $\mathcal{Y}^{\mathcal{A}}$. The abstract question $\mathcal{X}^{\mathcal{A}}$ is also fed into the oracle LLM, to complement the problem contexts and clarify the abstract symbols of input variables. Given the distilled abstract answer $\mathcal{Y}^{\mathcal{A}}$, we then conduct the abstraction retrieval to get the de-contextualized abstraction $\mathcal{A}$. Finally, the symbolic derivation step is performed to verify whether $\mathcal{A}$ along with $\mathcal{C}$ can derive the correct final answer stated in $\mathcal{Y}$. We only keep the rewritten instances that pass the answer verification.

### 3.2    LEARNING OF ABSTRACT REASONING

**Motivations**    Previous study (Gao et al., 2025) has shown that LLMs are poor at following in-context demonstrations to reason in abstract manner, indicating that the learning of abstract reasoning requires training with proper supervision, rather than relying on only in-context instructions and examples. A straightforward way is to train LLMs with supervised fine-tuning (SFT). However,

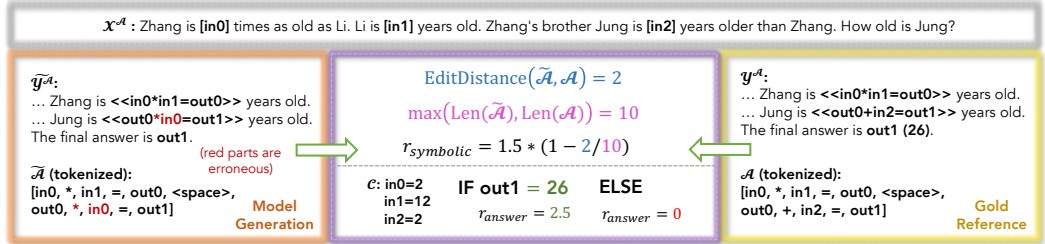

Figure 5: Illustration of the abstraction rewards in our reinforcement learning approach, including the symbolic distance reward $r_{symbolic}$ and the answer correctness reward $r_{answer}$.

although SFT on abstract data can teach LLMs decent abstract reasoning formats, its auto-regressive training objective also forces LLMs to learn the specific contexts of each training sample. This hinders LLMs from learning more general abstract thinking strategy, which leads to frequent test-time failure of generating an abstraction that is aligned with the problem, skewed by the new contexts in the testing data[3], as shown by our results in §5.1. Therefore, we propose to conduct reinforcement learning (RL) on top of SFT, to augment LLMs' capability of constructing faithful abstractions.

**Supervised Fine-Tuning** We fine-tune LLMs to auto-regressively generate our constructed GranulAR answer $\mathcal{Y}^{\mathcal{A}}$ based on the input question $\mathcal{X}^{\mathcal{A}}$, simply with the causal language modeling loss of predicting each token in $\mathcal{Y}^{\mathcal{A}}$ based on former tokens.

**Reinforcement Learning with Abstraction Rewards** On top of SFT, we develop a RL approach to further improve the faithfulness of abstraction generated by LLMs. Our RL approach proposes a new set of rewards to closely rate the generated abstraction on two aspects. First, given the model created abstraction $\widetilde{\mathcal{A}}$ retrieved from its generated answer $\widetilde{\mathcal{Y}^{\mathcal{A}}}$, we perform the symbolic derivation to check whether $\widetilde{\mathcal{A}}$ can derive the correct final answer (denoted as Ans) with the conditions $\mathcal{C}$ given in the gold reference. If yes, a positive reward $r_{correct}$ (hyperparameter) is granted to the model, otherwise zero reward is given. We denote this **answer correctness** reward as $r_{answer}(\widetilde{\mathcal{A}}, \mathcal{C}, \text{Ans})$. Second, we more granularly measure how $\widetilde{\mathcal{A}}$ is aligned with the expected abstraction $\mathcal{A}$ retrieved from the gold answer $\mathcal{Y}^{\mathcal{A}}$. Specifically, we split $\widetilde{\mathcal{A}}$ and $\mathcal{A}$ into lists of symbolic tokens, where each token is either an abstract symbol (created in $C$, $\widetilde{\mathcal{Y}^{\mathcal{A}}}$ or $\mathcal{Y}^{\mathcal{A}}$) that represents an input or output variable (such as in0), or a pre-defined operator that connects variables (such as =) or separates derivations (such as $\langle$space$\rangle$), as shown in Figure 5. Based on that, we calculate a **symbolic distance** reward:

$$r_{symbolic}(\widetilde{\mathcal{A}}, \mathcal{A}) = r_{max} \cdot \left(1 - \text{EditDistance}(\widetilde{\mathcal{A}}, \mathcal{A}) \Big/ \max_{a \in \{\widetilde{\mathcal{A}}, \mathcal{A}\}} \text{Len}(a)\right) \qquad (1)$$

where $r_{max}$ denotes the maximum reward hyperparameter, $\text{EditDistance}(\widetilde{\mathcal{A}}, \mathcal{A})$ denotes the list-wise edit distance (Levenshtein et al., 1966) between tokenized abstractions $\widetilde{\mathcal{A}}$ and $\mathcal{A}$, and $\text{Len}(\cdot)$ calculates the length of a list, used for normalizing the edit distance into the range of 0 to 1. A higher symbolic distance reward is granted to the model if $\widetilde{\mathcal{A}}$ is closer (or more similar) to $\mathcal{A}$, which gives the model more fine-grained learning signal of how far away it is from creating the correct abstraction. Figure 5 shows an example case of our abstraction rewards. Note that our proposed rewards do not require a pre-trained reward model, but just comparison to the gold reference. We plug our abstraction rewards with GRPO (Shao et al., 2024), an advanced RL algorithm for incentivizing reasoning capability in LLMs (Guo et al., 2025). The formulation of GRPO with our rewards is described in Appendix A.

## 4 EXPERIMENTAL SETTINGS

**Implementation of AbstRaL Framework** We prompt a Llama-3.3-70B-Instruct (Grattafiori et al., 2024) model with few-shot examples to accomplish the first **condition recognition** step of AbstRaL.

---

[3]On the other hand, learning to directly generate de-contextualized abstraction is rather hard for LLMs that are pre-trained on mostly contextualized natural language corpus, as verified by our analysis in §5.1, which motivates our use of a fine-grained framework to still incorporate contexts in the learning of abstract reasoning.

Based on that, we tune various LLMs on our constructed GranulAR data (§3.1), with our SFT and RL scheme (§3.2), to perform the core **abstract reasoning** step. Our tested LLMs include Llama3 model series (Grattafiori et al., 2024), Qwen2.5 model series (Yang et al., 2024a) and Mathstral-7B. In the **abstraction retrieval** step, we simply use a regex-matching script to extract all math derivations that are enclosed in double angle brackets in the abstract answer, which form the abstraction. We use SymPy equation solver to perform the **symbolic derivation**, which derives the final answer. For GranulAR training **data construction**, we use Llama-3.3-70B-Instruct as the oracle LLM, and prompt it with few-shot examples to rewrite the training samples in the socratic version of GSM8K (Cobbe et al., 2021) from OpenAI. More implementation details are included in Appendix B.

**Baseline Methods**    We compare AbstRaL to several baseline reasoning methods, including 8-shot CoT prompting (**CoT-8S**), 8-shot Program-of-Thought (Chen et al., 2022) prompting (**PoT-8S**), RL with non-rewritten (socratic CoT[4]) GSM8K training data (**CoT-RL**), and other abstract reasoning methods **CoA** (Gao et al., 2025), **AoT** (Hong et al., 2024) and **SyReLM** (Dutta et al., 2024). We include more details of baseline methods in Appendix C.

**Evaluation Datasets**    We evaluate our method on two datasets that are commonly used for testing GSM reasoning robustness: GSM-Symbolic (Mirzadeh et al., 2025) and GSM-Plus (Li et al., 2024). GSM-Symbolic manually constructs problem templates from 100 GSM8K testing samples, and uses the templates to create new problems where the numbers or names or both in the original problem are varied to different values, denoted as **Vary Num.**, **Vary Name** and **Vary Both**, respectively. We follow GSM-Symbolic to conduct 50 rounds of evaluation, each round creating 1 problem from each template (so models are tested on 100 new problems per round), and measure the average performance across these 50 rounds, *i.e.*, the mean (with standard deviation) of accuracy, and check whether it matches the performance on the original 100 (**Origin 100**) GSM8K problems. GSM-Plus creates different variations of the full GSM8K testing set instead, where each varied testing set contains all 1319 GSM8K testing problems. For each type of variation, 1 varied sample is created for each original problem, which leads to a single round of evaluation per variation. Our tested GSM-Plus variations include digit expansion (**Digit Ex.**) that adds more digits to a number (*e.g.*, from 16 to 1600), integer-decimal-fraction conversion (**Int-Dec-Fra**) that changes the type of a number (*e.g.*, from 2 to 2.5), numerical substitution (**Num. Sub.**) that replaces a number with another same-digit one (*e.g.*, from 16 to 20), rephrasing the question to check problem understanding (**Rephrase**), and distractor insertion (**Distract**) that adds topic-related but useless conditions, compared with model performance on the **Original** GSM8K testing set.

## 5    EXPERIMENTAL RESULTS

### 5.1    GSM REASONING ROBUSTNESS

Table 1 shows some of our representative evaluation results on GSM-Symbolic and GSM-Plus datasets. We report the performances of the smallest and the strongest LLMs tested in our experiments, which are Qwen2.5-0.5B-Instruct and Qwen2.5-Math-7B-Instruct, respectively. Results on other LLMs and on some data portions of GSM-Symbolic (Vary Num. and Vary Name) and GSM-Plus (Digit Ex., Int-Dec-Fra and Num. Sub.) are included in Appendix D, which draw the same conclusion.

On GSM-Symbolic, we find that AbstRaL effectively improves the reasoning robustness of all tested LLMs ranging from 0.5B to 7B (or 8B) sizes, with respect to the variations of both numbers and names (Vary Both), demonstrating better generalization to distribution shifts. Specifically, compared to baseline methods, LLMs with AbstRaL achieve consistently better (mean) accuracy on Vary Both samples, with overall lower standard deviation across different testing rounds. Besides, LLMs with AbstRaL suffer less performance drop ($\Delta$) when transferring from Origin 100 to Vary Both, especially compared to CoT and PoT methods that do not use any abstract reasoning. Interestingly, on some large-size (7B or 8B) LLMs, although AbstRaL scores lower than CoT-8S and CoT-RL on Origin 100, it outperforms these CoT methods on Vary Both perturbed samples. This implies that learning with AbstRaL may mitigate LLMs' overfitting to the existing input conditions, caused by potential data contamination (Magar & Schwartz, 2022; Xu et al., 2024) at pre-training or post-training stage.

---

[4]We also did a pilot study of training LLMs with non-socratic CoT data, which achieves similar results.

Table 1: Evaluation results of GSM reasoning robustness, measured by the accuracy (%) of final answer. $\Delta$ denotes the relative percentage of drop comparing performance on **Vary Both** to performance on **Origin 100**. **Num. Pert.** denotes the average performance on the three GSM-Plus testing sets that perturb input numbers (i.e., Digit Ex., Int-Dec-Fra and Num. Sub.). Best results on each model are **bold**, where lower is better for $\Delta$. Standard deviation (std) of multi-round evaluation results on **Vary Both** are in brackets, where lowest std on each model are underlined.

| Model | Method | GSM-Symbolic | | | GSM-Plus | | | |
|---|---|---|---|---|---|---|---|---|
| | | Vary Both | Origin 100 | $\Delta$ | Num. Pert. | Rephrase | Distract | Original |
| Qwen2.5-0.5B-Instruct | CoT-8S | 34.0 (0.033) | 38.0 | 10.6 | 34.8 | 43.6 | 22.7 | 42.4 |
| | PoT-8S | 27.0 (0.035) | 31.0 | 12.8 | 28.7 | 36.3 | 19.4 | 35.5 |
| | CoT-RL | 32.3 (0.032) | 35.0 | 7.77 | 30.9 | 40.8 | 15.2 | 38.0 |
| | CoA | 28.7 (0.026) | 29.0 | 0.97 | 26.0 | 31.6 | 12.7 | 30.3 |
| | AoT | 27.8 (0.029) | 29.0 | 4.21 | 25.3 | 32.1 | 12.7 | 30.7 |
| | SyReLM | 36.8 (0.030) | 39.0 | 5.54 | 38.7 | 41.6 | 21.1 | 41.5 |
| | AbstRaL | **44.6** (0.025) | **44.0** | **-1.27** | **46.7** | **46.3** | **36.5** | **46.7** |
| Qwen2.5-Math-7B-Instruct | CoT-8S | 87.7 (0.023) | 95.0 | 7.73 | 84.1 | **90.5** | 76.3 | 91.8 |
| | PoT-8S | 86.7 (0.023) | 91.0 | 4.77 | 82.8 | 87.9 | 75.1 | 87.6 |
| | CoT-RL | 88.1 (0.021) | **96.0** | 8.21 | 86.0 | 90.4 | 81.5 | **93.4** |
| | CoA | 69.7 (0.033) | 71.0 | 1.83 | 71.8 | 76.4 | 57.1 | 78.1 |
| | AoT | 70.4 (0.032) | 72.0 | 2.17 | 71.7 | 76.7 | 56.6 | 77.9 |
| | SyReLM | 87.7 (0.026) | 90.0 | 2.56 | 84.4 | 88.6 | 76.0 | 88.6 |
| | AbstRaL | **90.2** (0.016) | 91.0 | **0.86** | **89.1** | 89.9 | **82.3** | 89.2 |

Similarly on GSM-Plus, AbstRaL almost reverts the performance degradation caused by variations of input numbers (Num. Pert.), and also maintains robustness to contextual variations (Rephrase) comparable to baseline methods. More interestingly, AbstRaL significantly mitigates the interference of distracting conditions added to the problems (Distract), while by comparison, LLMs with baseline reasoning methods all score more drastically lower when transferring from Original to Distract. AbstRaL's improvement on Distract is largely due to learning on its granularly-decomposed abstract reasoning (GranulAR) data (verified in our ablation study below), which enables overall planning of the reasoning steps and then rethinking of useful input conditions at each reasoning step.

**Ablation Study**    One natural concern of our method is whether the improvements of AbstRaL are due to the learning of abstract reasoning, or just because of integrating powerful symbolic tools, *i.e.*, Llama-3.3-70B-Instruct prompted for condition recognition, regex-matching script for abstraction retrieval and SymPy equation solver for symbolic derivation. To clarify this, we conduct an ablation study (**w/o Tools**), which replaces all above symbolic tools in our framework with the LLM itself, to accomplish each step of our pipeline with step-specific prompts. Results in Table 2 show that LLMs without additional tools suffer only minor performance drop, indicating that the learning of abstract reasoning in our framework is the major contributor to the improvements.

We also investigate the effect of using AbstRaL's pipeline to construct the abstraction within contexts, *i.e.*, $\mathcal{X} \rightarrow \mathcal{X}^{\mathcal{A}} \rightarrow \mathcal{Y}^{\mathcal{A}} \rightarrow \mathcal{A}$, by testing an ablated framework (**w/o Contexts**), where LLMs are trained (with SFT and the same RL approach) to directly generate the abstraction $\mathcal{A}$ based on the abstract input question $\mathcal{X}^{\mathcal{A}}$, *i.e.*, $\mathcal{X} \rightarrow \mathcal{X}^{\mathcal{A}} \rightarrow \mathcal{A}$ without explicitly generating $\mathcal{Y}^{\mathcal{A}}$ and post-processing it by abstraction retrieval. As shown in Table 2, LLMs that learn inferences without intermediate contexts fall far behind the unablated ones. This indicates that narrowing down the gap between symbolic abstraction and natural language is important to faithful abstract thinking.

Our third ablation studies the contribution of RL in AbstRaL, by totally ablating the RL (**w/o RL**) or removing only the symbolic distance reward (**w/o** $r_{symbolic}$). On both the original and perturbed testing sets, we find that LLMs without RL score far lower than reinforcement-learned ones. Without $r_{symbolic}$ that hints how close the generated abstraction is to the correct one, LLMs also suffer significant performance drop, and thus under-perform the prompting baseline (CoT-8S). These findings demonstrate that acquiring faithful abstract reasoning requires careful learning, *e.g.*, via a proper RL approach and a milestone-style reward that closely monitors the progress of learning.

Lastly, we ablate the granularly-decomposed abstract reasoning (**w/o GranulAR**) used as AbstRaL's training data format. We alter the gold abstract answer $\mathcal{Y}^{\mathcal{A}}$ in our training data back to the standard socratic CoT format (as shown in Figure 4), where we remove listing sub-questions at the start of

Table 2: Ablation study results of GSM reasoning robustness.

| Model | Method | GSM-Symbolic | | | GSM-Plus | | | |
|---|---|---|---|---|---|---|---|---|
| | | Vary Both | Origin 100 | Δ | Num. Pert. | Rephrase | Distract | Original |
| Qwen2.5-0.5B-Instruct | AbstRaL | 44.6 | 44.0 | -1.27 | 46.7 | 46.3 | 36.5 | 46.7 |
| | - w/o Tools | 43.6 | 44.0 | 0.86 | 46.0 | 45.6 | 35.6 | 46.1 |
| | - w/o Contexts | 23.1 | 30.0 | 23.1 | 27.0 | 28.8 | 16.5 | 28.7 |
| | - w/o RL | 29.6 | 36.0 | 17.9 | 38.0 | 32.3 | 28.3 | 35.7 |
| | - w/o $r_{symbolic}$ | 37.6 | 39.0 | 3.59 | 41.7 | 40.0 | 31.3 | 38.8 |
| | - w/o GranulAR | 42.6 | 42.0 | -1.48 | 42.4 | 42.3 | 23.6 | 42.4 |
| Qwen2.5-Math-7B-Instruct | AbstRaL | 90.2 | 91.0 | 0.86 | 89.1 | 89.9 | 82.3 | 89.2 |
| | - w/o Tools | 90.0 | 91.0 | 1.14 | 88.6 | 90.4 | 81.8 | 88.9 |
| | - w/o Contexts | 69.3 | 71.0 | 2.45 | 67.2 | 69.4 | 55.1 | 70.7 |
| | - w/o RL | 78.5 | 82.0 | 4.22 | 82.1 | 80.0 | 74.5 | 82.3 |
| | - w/o $r_{symbolic}$ | 83.2 | 87.0 | 4.34 | 85.8 | 82.3 | 78.1 | 85.9 |
| | - w/o GranulAR | 88.1 | 89.0 | 0.97 | 88.4 | 89.8 | 66.8 | 88.6 |

Table 3: Zero-shot evaluation results on OOD math datasets. Best results on each model are **bold**.

| Model | GSM Train Method | MATH | Minerva MATH | SAT-Math | AIME-24 | SVAMP | ASDiv | MAWPS | Gaokao-2023 | TABMWP | College-Math | AQUA |
|---|---|---|---|---|---|---|---|---|---|---|---|---|
| Qwen2.5-0.5B -Instruct | Ori-SFT | 30.0 | 5.5 | 46.9 | 0.0 | 56.0 | 74.3 | 88.1 | 27.0 | 48.8 | 26.2 | 31.5 |
| | CoT-RL | 19.2 | 5.5 | 43.8 | 0.0 | 45.2 | 64.2 | 84.4 | 16.6 | 33.1 | 16.2 | 26.4 |
| | CoA | 31.6 | 5.5 | 43.8 | 0.0 | 39.8 | 54.0 | 60.1 | 23.9 | 46.2 | 27.8 | 30.7 |
| | AbstRaL | **34.7** | **7.4** | **62.5** | 0.0 | **62.7** | **77.4** | **89.8** | **29.1** | **50.1** | **28.5** | **32.7** |
| Qwen2.5-Math -7B-Instruct | Ori-SFT | 83.5 | 34.6 | 90.6 | 13.3 | 93.5 | **95.3** | 98.5 | 67.5 | **93.9** | 47.1 | 74.4 |
| | CoT-RL | 82.9 | 35.7 | 89.8 | 10.0 | 92.5 | 94.9 | 98.1 | 67.5 | 89.2 | 46.8 | 65.4 |
| | CoA | 83.7 | 34.6 | 90.0 | 12.7 | 93.1 | 95.1 | 98.3 | 67.0 | 92.9 | **47.2** | 72.8 |
| | AbstRaL | **83.9** | **38.6** | **93.8** | **16.7** | **94.3** | **95.3** | **98.5** | **68.3** | **93.9** | **47.2** | **74.8** |

reasoning chain and quoting of input conditions at each reasoning step, while just keep the abstract derivations that forms the abstraction $\mathcal{A}$. LLMs trained on this ablated data suffer drastic performance drop on the Distract testing set, indicating that they are vulnerable to useless distractors. This verifies that our adopted fine-grained reasoning format plays an essential role in identifying interference conditions, due to planning the reasoning steps and useful input conditions in each step.

## 5.2 GENERALIZATION TO OOD TASKS

AbstRaL's learning of abstract thinking in GSM problems also improves general reasoning capabilities of LLMs, rather than overfitting LLMs to GSM-specific tasks. We verify this by testing LLMs' zero-shot generalizability on a wide range of OOD datasets beyond GSM8K, including other math datasets MATH (Hendrycks et al., 2021), Minerva MATH (Lewkowycz et al., 2022), SAT-Math from AGIEval (Zhong et al., 2024b), AIME24 (MAA, 2024), SVAMP (Patel et al., 2021), ASDiv (Miao et al., 2020), MAWPS (Koncel-Kedziorski et al., 2016), Gaokao-2023 (Liao et al., 2024), TABMWP (Lu et al., 2022), College-Math (Tang et al., 2024) and AQUA (Ling et al., 2017), and general reasoning datasets MMLU (Hendrycks et al., 2020), BBH (Suzgun et al., 2023), ARC-Challenge (Clark et al., 2018) and OpenBookQA (Mihaylov et al., 2018). We compare Qwen models trained with AbstRaL to the original model checkpoints that are already fine-tuned on GSM8K in post-training, denoted as **Ori-SFT**. We also compare AbstRaL to the baseline method CoT-RL, which improves the original GSM8K performance of Qwen2.5-Math-7B-Instruct, and to the baseline CoA, which achieves second-best GSM reasoning robustness (based on the ranking of Δ).

Table 3 and 4 present our zero-shot generalization results on OOD math and general reasoning datasets, respectively. AbstRaL consistently outperforms baseline methods across all the tested OOD datasets, even though the GSM reasoning scheme learned in AbstRaL training may not be directly applicable to solving the OOD tasks. This indicates that learning the abstract thinking underlying GSM can already **implicitly benefit** other mathematical reasoning and more general reasoning of LLMs. By contrast, Qwen models trained with CoT-RL and CoA cannot stably improve upon the original checkpoints (Ori-SFT), showing that acquiring such "implicit benefit" from GSM needs more careful learning via AbstRaL.

Table 4: Zero-shot evaluation results on other OOD datasets. Best results on each model are **bold**.

| Model | GSM Train Method | MMLU | | | | BBH | ARC-Challenge | OpenBook-QA |
|---|---|---|---|---|---|---|---|---|
| | | STEM | Social | Humanities | Other | | | |
| Qwen2.5-0.5B-Instruct | Ori-SFT | 37.5 | 51.5 | 41.6 | 49.7 | 21.0 | 33.3 | 33.6 |
| | CoT-RL | 38.5 | 50.8 | 41.2 | 49.1 | 23.2 | 32.2 | 33.2 |
| | CoA | 38.5 | 50.8 | 41.9 | 49.3 | 21.8 | 32.6 | 33.6 |
| | AbstRaL | **38.8** | **53.5** | **42.2** | **50.7** | **26.3** | **33.8** | **34.6** |
| Qwen2.5-Math-7B-Instruct | Ori-SFT | 68.4 | 82.5 | 63.2 | 76.4 | 43.7 | 51.9 | 48.8 |
| | CoT-RL | 68.5 | 81.5 | 63.0 | 74.4 | 46.0 | 52.0 | 47.0 |
| | CoA | 68.5 | 81.7 | 63.3 | 75.5 | 42.9 | 51.0 | 48.4 |
| | AbstRaL | **68.6** | **82.7** | **63.8** | **76.5** | 54.8 | 52.3 | **49.4** |

## 5.3 DISCUSSION

**Theoretically, why AbstRaL leads to better reasoning robustness and generalizability?** The abstractions studied in AbstRaL are high-level inference structures that serve as scaffolds for reasoning, and robust generalization of LLM reasoning requires understanding of such abstractions (Chen et al., 2025a; Saitta et al., 2013), which is improved by AbstRaL's learning. Abstract thinking is also an essential component of general fluid intelligence (Chollet, 2019), and AbstRaL's reinforcement of abstract thinking also improves such intelligence of LLMs, which naturally contributes to better generalizability of LLM reasoning. In line with the globality degree (Abbe et al., 2024) theory, AbstRaL sets an intermediate learning step (the problem abstraction) that is likely useful to reach the final step (answering the question). This may narrow down the globality degree, i.e., distribution gap between the input and the output of the reasoning task, and reduce the chance of modeling shallow reasoning patterns or shortcuts. It would be an interesting future work to establish these formally.

**Whether the improvements of AbstRaL simply rely on distilling superior reasoning capabilities from the oracle LLM?** We use an oracle LLM (Llama-3.3-70B-Instruct) to perform the input condition recognition and our GranulAR training data construction. However, AbstRaL does not necessarily rely on this strong oracle LLM, and a weaker LLM can be used as the oracle instead.

We clarify that both input condition recognition and GranulAR data construction in AbstRaL are based on re-writing the existing Socratic CoT data. The oracle LLM only serves to re-write the surface-form format of existing reasoning traces, *i.e.*, replacing real numerical values with abstract symbols and re-arranging the decomposed sub-question answering. Therefore, the improvements of our framework are not due to distilling more superior reasoning knowledge from the oracle LLM, but due to the AbstRaL learning scheme itself.

We have verified in our ablation study (Table 2 - w/o Tools) that replacing the oracle LLM with the reasoning LLM itself to conduct input condition recognition results in only minor performance drop. Furthermore, we have tested to use a weaker LLM, Llama-3.1-8B-Instruct, to re-write our GranulAR reasoning traces and conduct AbstRaL training based on that, which achieves roughly the same scores on our tested GSM perturbation and OOD datasets. These justify that it's mainly the better learning of abstract reasoning that contributes to the improvements of AbstRaL.

**Why LLMs, which are also trained on code and structured languages besides natural language corpora, still fall short of predicting de-contextualized abstractions?** In our ablation study (Table 2 - w/o Contexts), we have verified that it is still hard for LLMs to generate GSM abstractions without intermediate contexts. We assume that this is because most reasoning data samples in LLM pre-training and post-training corpora are still in natural language format (*e.g.*, CoT traces), and thus LLMs might fall short of integrating multi-step reasoning capabilities within the abstract symbolic format, which may be less frequently seen in the reasoning training data samples.

## 6 RELATED WORK

**Reasoning Robustness** Recent advances in LLM reasoning have also spotted considerable robustness challenges (Yu et al., 2025), reflected by benchmarks in symbolic reasoning, such as logical (Bao et al., 2023) and mathematical (Sundaram et al., 2024; Li et al., 2024; Srivastava et al., 2024;

Mirzadeh et al., 2025) reasoning, and in factual or commonsense reasoning (Yang et al., 2023; Qian et al., 2024; Qiang et al., 2024). All above benchmarks consistently reveal that LLMs are easily perturbed by test-time data distribution shifts. Inspired by improving robustness via data augmentation (Rebuffi et al., 2021b), previous works (Gan & Ng, 2019; Rebuffi et al., 2021a; Zhou et al., 2024; Boix-Adserà et al., 2024) use various data synthesis techniques to expand the coverage of training samples, and thus anticipate potential distribution shifts, which naturally increases the computational cost of developing a LLM. In this work, we aim to improve reasoning robustness of LLMs by incentivizing their abstract thinking, instead of replying on larger amount of instantiations.

**Abstract Thinking and Planning** Abstract thinking is an essential component of general fluid intelligence (Chollet, 2019), and is also the key to human cognition and reasoning (Saitta et al., 2013). It requires making inferences based on abstract fundamental rules or concepts (Chen et al., 2025a), rather than just memorizing a probabilistic pattern matching (Gendron et al., 2024; Xu et al., 2025). Recently, various reasoning (or data) formats have been proposed for LLMs to learn abstract thinking, such as AoT (Hong et al., 2024) and CoA (Gao et al., 2025) based on natural language, and PoT (Chen et al., 2022), PAL (Gao et al., 2023) and SyReLM (Dutta et al., 2024) based on programming or formal language. On the other hand, planning is also a basic human reasoning skill (Wilensky, 1983), which benefits LLM reasoning. Typical planning methods, such as chain-of-thought (CoT) (Wei et al., 2022) and socratic problem decomposition (Shridhar et al., 2023), are widely adopted in LLMs to improve reasoning. Our work develops a better learning scheme of abstract thinking based on more commonly used natural language, and also integrates it within the power of planning.

**Reinforcement Learning** RL is a popular technique used in recent development of LLMs (Shao et al., 2024; Guo et al., 2025), to boost reasoning capabilities. A representative RL approach is PPO (Schulman et al., 2017), typically used for learning from human feedback (RLHF) (Christiano et al., 2017; Ouyang et al., 2022) via a reward model pre-trained on human preference annotations. As a step forward, DPO (Rafailov et al., 2023) simplifies the PPO implementation by optimizing the direct preference of policy model, which gets rid of pre-training an additional reward model. Our method adopts GRPO (Shao et al., 2024) with model-free rewards, which further cuts off the training of value model (used for advantage estimation) by using a group relative advantage.

## 7 CONCLUSION

This paper proposes a method, AbstRaL, to promote abstract thinking in large language models (LLMs). AbstRaL is designed to improve GSM reasoning robustness of LLMs, based on the natural principle that abstract thinking leads to reasoning steps that are more invariant to surface-form variations. Our abstraction mechanism is implemented through a proper reinforcement learning (RL) framework, where model-free rewards are derived from newly designed rationales GranulAR that blend socratic chain-of-thought with augmented granularity. This enables both the decontextualization of problems and the integration of symbolic tools. We evaluate AbstRaL on recent GSM perturbation benchmarks and show that it effectively mitigates the performance degradation caused by instantiation and interferential shifts. Our zero-shot evaluation on OOD tasks further demonstrates AbstRaL's effectiveness on improving LLMs' general reasoning capabilities, which implies potential to extend our learning method to a broader range of domains in future work.

## ACKNOWLEDGEMENTS

Antoine Bosselut gratefully acknowledges the support of the Swiss National Science Foundation (No. 215390), Innosuisse (PFFS-21-29), the EPFL Center for Imaging, Sony Group Corporation, a Meta LLM Evaluation Research Grant, and the Swiss National Supercomputing Center (CSCS) in the form of an infrastructure engineering and development project.

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

# A  GRPO WITH ABSTRACTION REWARDS

Our proposed AbstRaL framework adopts GRPO (Shao et al., 2024) as the RL algorithm to train LLMs (on top of supervised fine-tuning) on the task of abstract reasoning, *i.e.*, generating the abstract answer $\mathcal{Y}^{\mathcal{A}}$ based on the input abstract question $\mathcal{X}^{\mathcal{A}}$.

For each input question $\mathcal{X}^{\mathcal{A}}$, GRPO samples a group of output answers $\{\widetilde{\mathcal{Y}_1^{\mathcal{A}}}, \widetilde{\mathcal{Y}_2^{\mathcal{A}}}, ..., \widetilde{\mathcal{Y}_G^{\mathcal{A}}}\}$ from the current (old policy) model $\pi_{\theta_{old}}$, and optimizes the (policy) model $\pi_\theta$ by maximizing the objective:

$$\frac{1}{G} \sum_{i=1}^{G} \left( \min \left( \frac{\pi_\theta\left(\widetilde{\mathcal{Y}_i^{\mathcal{A}}}|\mathcal{X}^{\mathcal{A}}\right)}{\pi_{\theta_{old}}\left(\widetilde{\mathcal{Y}_i^{\mathcal{A}}}|\mathcal{X}^{\mathcal{A}}\right)} R_i, \text{clip}\left( \frac{\pi_\theta\left(\widetilde{\mathcal{Y}_i^{\mathcal{A}}}|\mathcal{X}^{\mathcal{A}}\right)}{\pi_{\theta_{old}}\left(\widetilde{\mathcal{Y}_i^{\mathcal{A}}}|\mathcal{X}^{\mathcal{A}}\right)}, 1-\varepsilon, 1+\varepsilon \right) R_i \right) - \beta \mathbb{D}_{KL}\left(\pi_\theta\|\pi_{ref}\right) \right) \quad (2)$$

where $\varepsilon$ and $\beta$ are hyperparameters. We set the reference policy $\pi_{ref}$ as the model trained with only supervised fine-tuning (SFT), which is used for calculating the KL divergence regularization:

$$\mathbb{D}_{KL}\left(\pi_\theta\|\pi_{ref}\right) = \frac{\pi_{ref}\left(\widetilde{\mathcal{Y}_i^{\mathcal{A}}}|\mathcal{X}^{\mathcal{A}}\right)}{\pi_\theta\left(\widetilde{\mathcal{Y}_i^{\mathcal{A}}}|\mathcal{X}^{\mathcal{A}}\right)} - \log \frac{\pi_{ref}\left(\widetilde{\mathcal{Y}_i^{\mathcal{A}}}|\mathcal{X}^{\mathcal{A}}\right)}{\pi_\theta\left(\widetilde{\mathcal{Y}_i^{\mathcal{A}}}|\mathcal{X}^{\mathcal{A}}\right)} - 1 \quad (3)$$

$R_i$ is the group relative advantage granted to the abstraction $\widetilde{\mathcal{A}}_i$ retrieved from each sampled output answer $\widetilde{\mathcal{Y}_i^{\mathcal{A}}}$, which applies a group normalization on our proposed abstraction rewards $r_{answer}$ and $r_{symbolic}$ defined in §3.2, with reference to the input conditions $\mathcal{C}$ and gold answer Ans, and to the gold abstraction $\mathcal{A}$ retrieved from the gold response $\mathcal{Y}^{\mathcal{A}}$, respectively:

$$R_i = \frac{r_i - \text{mean}(\{r_1, r_2, \cdots, r_G\})}{\text{std}(\{r_1, r_2, \cdots, r_G\})}, \quad r_i = r_{answer}(\widetilde{\mathcal{A}}_i, \mathcal{C}, \text{Ans}) + r_{symbolic}(\widetilde{\mathcal{A}}_i, \mathcal{A}) \quad (4)$$

# B  IMPLEMENTATION DETAILS OF ABSTRAL

**Condition Recognition**  We create few-shot examples to prompt a Llama-3.3-70B-Instruct (Grattafiori et al., 2024) model to accomplish the first condition recognition step of AbstRaL. Table 10 presents the instruction and few-shot examples used as the task demonstration in our prompting. The LLM is tasked to label the numerical values in the input question $\mathcal{X}$ with square brackets "[]", and then sequentially replace each labeled value with an abstract symbol indexing as "in0", "in1", etc., to create the abstract question $\mathcal{X}^{\mathcal{A}}$. Note that for implicit numerical values such as "one hundred" and "twice", we also prompt the LLM to convert them into explicit format "100" and "2 times", so that the numbers can be labeled and replaced. Meanwhile, the LLM is asked to use equations to record the replacements, *e.g.*, "in0 = 2", in order to create the conditions $\mathcal{C}$. 4 NVIDIA A100-SXM4 (80GB) GPUs were used for running the condition recognition step based on Llama-3.3-70B-Instruct, which took about 36 hours to process all training and testing data samples.

**Abstract Reasoning**  We tune various LLMs on our constructed GranulAR data (§3.1), with our SFT and RL scheme (§3.2), to perform the core **abstract reasoning** step. Our tested LLMs include Llama3 model series (Grattafiori et al., 2024) (Llama-3.2-1B-Instruct, Llama-3.2-3B-Instruct and Llama-3.1-8B-Instruct), Qwen2.5 model series (Yang et al., 2024a) (Qwen2.5-0.5B-Instruct, Qwen2.5-1.5B-Instruct, Qwen2.5-3B-Instruct, Qwen2.5-7B-Instruct and its math specialized version Qwen2.5-Math-7B-Instruct) and Mathstral-7B[5]. All LLMs are supposed to follow our GranulAR training data to index the derived output variables in their generations as "out0", "out1", etc., and highlight each abstract math derivation in double angle brackets such as "$\langle\langle$in0 * in1 = out0$\rangle\rangle$". LLMs are also supposed to generate a fixed statement "The final answer is [outN].", to clarify the output variable that represents the final answer. At inference phase, LLMs use greedy decoding to generate their abstract answers.

For SFT, we set the batch size as 8, using 4 NVIDIA A100-SXM4 (80GB) GPUs (*i.e.*, batch size is 2 on each GPU), and set the learning rate as $5e^{-6}$, using AdamW Loshchilov & Hutter (2018) optimizer with $\beta_1 = 0.9$, $\beta_2 = 0.999$ and $\epsilon = 1e^{-8}$. All LLMs were trained with SFT for 2 epochs on our GranulAR data, which took less than 1 hour.

For RL, we set the positive (correct) reward $r_{correct} = 2.5$ in our answer correctness reward $r_{answer}(\widetilde{\mathcal{A}}_i, \mathcal{C}, \text{Ans})$, and the maximum reward $r_{max} = 1.5$ in our symbolic distance reward

---

[5] https://huggingface.co/mistralai/Mathstral-7B-v0.1

$r_{symbolic}(\widetilde{\mathcal{A}}_i, \mathcal{A})$. The hyperparameters of GRPO algorithm are set to $\beta = 0.04$ (KL coefficient) and $\epsilon = 0.2$ (for clipping), with number of generations (*i.e.*, $\{\widetilde{\mathcal{Y}_1^{\mathcal{A}}}, \widetilde{\mathcal{Y}_2^{\mathcal{A}}}, ..., \widetilde{\mathcal{Y}_G^{\mathcal{A}}}\}$) sampled per group (*i.e.*, per given $\mathcal{X}^{\mathcal{A}}$) set as $G = 16$, and with the temperature, $\text{top}_p$ and $\text{top}_k$ of the sampling set as 0.9, 1.0 and 50, respectively. The learning rate of our RL optimization is set to $5e^{-7}$, using also AdamW optimizer with $\beta_1 = 0.9$, $\beta_2 = 0.99$ and $\epsilon = 1e^{-8}$. A cosine learning rate scheduler is employed with warm-up ratio set to 0.1, and the training weight decay and gradient norm clip value are also both set to 0.1. We use 8 NVIDIA A100-SXM4 (80GB) GPUs to train each LLM with RL. For 7B and 8B LLMs, the batch size of generation is set to 2 per GPU, with gradient accumulation steps set as 4, *i.e.*, the policy gradient update is performed when every 4 groups of $G = 16$ generations (2 batch size multiply 8 GPUs) are sampled. For other smaller LLMs, the batch size of generation is set to 8 per GPU instead, with gradient accumulation step set as 1, where still 4 groups of $G = 16$ generations (8 batch size multiply 8 GPUs) are sampled per policy gradient update. All LLMs were trained with RL for 8 epochs on our GranulAR data, which took about 3 to 5 days.

**Abstraction Retrieval and Symbolic Derivation**    We simply use a regex-matching script to extract all math derivations that are enclosed in double angle brackets in the model generated answer $\widetilde{\mathcal{Y}^{\mathcal{A}}}$ (or $\mathcal{Y}^{\mathcal{A}}$ in our GranulAR data construction). All extracted math derivations form the problem abstraction $\widetilde{\mathcal{A}}$ (or $\mathcal{A}$). During the tokenization of the abstraction (used for calculating the symbolic distance reward in our RL approach), the tokenized lists of derivations are concatenated with a special token "$\langle space \rangle$", as shown in Figure 5. We treat the output math derivations in $\widetilde{\mathcal{A}}$ (or $\mathcal{A}$) and the input conditions in $\mathcal{C}$ jointly as a system of equations, which is fed into a SymPy[6] equation solver to derive the final answer.

**GranulAR Training Data Construction**    Based on the abstract question $\mathcal{X}^{\mathcal{A}}$ and the gold socratic CoT response $\mathcal{Y}$ to the question, we prompt Llama-3.3-70B-Instruct to rewrite $\mathcal{Y}$ into our granularly-decomposed abstract reasoning (GranulAR) format $\mathcal{Y}^{\mathcal{A}}$, which is used as the training data for LLMs to learn the abstract reasoning step in AbstRaL. To facilitate the rewriting, we employ a two-step pipeline. First, the Llama model is prompted to replace the numerical values in $\mathcal{Y}$ with abstract symbols, by either quoting the abstract symbols in $\mathcal{X}^{\mathcal{A}}$ for input values or creating new abstract symbols for derived output values. All derivations in the response are supposed to be enclosed in double angle brackets "$\langle\langle\rangle\rangle$". Second, based on the rewritten response in the first step, the Llama model is prompted to further rewrite the response (in socratic CoT format) into our GranulAR format, while keep all the abstract symbols in the response unchanged. Table 11 and 12 present the instruction and few-shot examples used for prompting our two-step GranulAR training data construction. For each step of rewriting, 4 NVIDIA A100-SXM4 (80GB) GPUs were used to run the Llama model. About 36 and 48 hours are spent to conduct the first and the second rewriting steps, respectively, on all training data samples.

After each step of response rewriting, we filter the output of Llama model by verifying the correctness of its derivations, i.e., we use regex-matching to extract all derivations enclosed in "$\langle\langle\rangle\rangle$", and pass them (along with the input conditions $\mathcal{C}$ generated in the condition recognition step) into the SymPy solver, to derive the final answer (number), checking whether it matches the gold answer. We apply our two-step rewriting to the socratic version of GSM8K Cobbe et al. (2021) training set from OpenAI[7]. After the first step of rewriting and filtering, 6503 out of 7473 training samples are correctly rewritten and kept, while after the second step of rewriting and filtering, 6386 out of 6503 training samples are correctly rewritten and reserved as our final training data.

## C    DETAILS OF BASELINE METHODS

**CoT-8S** prompts LLMs with the demonstration template suggested by GSM-Symbolic (Mirzadeh et al., 2025) and the common 8-shot examples[8] used for GSM8K evaluation, to generate CoT answer to the input question. The last number (after "The final answer is") is extracted as the final answer.

---

[6] https://github.com/sympy/sympy
[7] https://huggingface.co/datasets/openai/gsm8k/tree/main/socratic
[8] https://github.com/EleutherAI/lm-evaluation-harness/blob/main/lm_eval/tasks/gsm8k/gsm8k-cot.yaml

**PoT-8S** (Chen et al., 2022) conducts GSM reasoning via Python programming. It prompts LLMs with 8-shot examples to generate program-based solution to the input problem, which is then executed by a Python interpreter to derive the final answer.

**CoT-RL** tunes LLMs on the non-rewritten GSM8K training data $(\mathcal{X}, \mathcal{Y})$ (socratic CoT version[9] with 7473 samples), using the same SFT and RL algorithm (GRPO) as our approach. Since CoT-RL does not generate abstractions in the reasoning chains, only our answer correctness reward $r_{answer}$ (§3.2) is used as the learning signal in RL, which checks whether the final answer (last number) extracted from the generated CoT matches the gold answer. In particular, $r_{answer} = r_{correct}$ (positive hyperparameter) if answers match, otherwise $r_{answer} = 0$.

**CoA** (Gao et al., 2025) is another abstract reasoning method that fine-tunes LLMs to generate abstract reasoning chains. However, different from AbstRaL, only the output numbers in CoA reasoning chains are represented by abstract symbols, without abstracting the input conditions ($\mathcal{X}^{\mathcal{A}}$ and $\mathcal{C}$) and using our granularly decomposed reasoning schema (GranulAR). The learning of CoA is based on only SFT, without integrating proper RL.

**AoT** (Hong et al., 2024) augments CoT data by adding abstract step descriptions before each step of the reasoning chains, and fine-tunes LLMs on the augmented data. Similar to the baseline method CoA, AoT trains LLMs via plain SFT, without RL that augments the faithfulness of abstract reasoning.

**SyReLM** (Dutta et al., 2024) similarly uses RL and symbolic rewards to augment the learning of abstract reasoning. However, different from AbstRaL, it formulates abstractions in programming or formal languages, which may hinder LLMs from integrating abstract reasoning with their pre-learned reasoning strategies (mostly in natural language) during pre-training or post-training.

## D    FULL EXPERIMENTAL RESULTS

Table 5 and 6 present the evaluation results of all our tested LLMs on GSM-Symbolic and GSM-Plus datasets, respectively. Results on all LLMs consistently demonstrate that AbstRaL effectively improves GSM reasoning robustness when generalizing to both instantiation and interferential shifts. On each tested LLM, we also conduct the bootstrap statistical significant test Xu (2006) between the best and second-best results, and highlight the best results (with *) if they are significantly better than their corresponding second-best results with significant test p-value $< 0.05$.

## E    QUALITATIVE ANALYSIS

Table 7 and 8 present two mathematical reasoning examples (respectively on GSM-Symbolic and GSM-Plus datasets) of our tested strongest LLM Qwen2.5-Math-7B-Instruct, using either baseline CoT-8S method or our AbstRaL. The LLM with AbstRaL performs more stable math derivations when facing the variations of relevant conditions (Table 7), and achieves more robust reasoning chains when dealing with the inserted distracting condition (Table 8). We include more detailed analysis in the corresponding table captions.

**Error Analysis**    AbstRaL teaches LLMs to analyze and decompose reasoning problems at the start of inference, which requires LLMs to perform more holistic problem understanding and solution planning, compared to CoT. This however may lead to slightly overthinking and thus misunderstanding of a few reasoning problems. We have manually checked the GSM8K testing questions where Qwen2.5-Math-7B-Instruct answered correctly with 8-shot CoT prompting (CoT-8S) but wrongly using AbstRaL (**totally 35** such questions), and verified that AbstRaL's failure on all these questions are due to minor misunderstanding. Table 9 presents an example where Qwen2.5-Math-7B-Instruct (AbstRaL) misunderstood "go up to" as "go up by", and therefore did a redundant reasoning step.

## F    THEORETICAL OPEN PROBLEMS

It would be interesting to find a theoretical setting to study the impact of abstraction on the sample/time complexity of learning, the OOD robustness and the model size requirements for learning. For

---

[9]We also did a pilot study of training LLMs with non-socratic CoT data, which achieves similar results.

Table 5: More evaluation results on GSM-Symbolic. $\Delta$ denotes the relative percentage of drop comparing performance on **Vary Both** to performance on **Origin 100**. Best results on each model are **bold**, where lower is better for $\Delta$. Standard deviation (std) of multi-round evaluation results are in brackets, where lowest std values on each model are underlined. The best (bold) results with $^*$ are significantly better than their corresponding second-best results, with significant test p-value $< 0.05$.

| Model | Method | Vary Num. | Vary Name | Vary Both | Origin 100 | $\Delta$ (%) |
|---|---|---|---|---|---|---|
| Llama-3.2-1B-Instruct | CoT-8S | 0.3864 (0.030) | 0.4250 (0.026) | 0.3890 (0.038) | 0.4800 | 18.96 |
| | CoT-RL | 0.4788 (0.031) | 0.5342 (0.024) | 0.4488 (0.038) | 0.5700 | 21.26 |
| | CoA | 0.4534 (0.020) | 0.4372 (0.026) | 0.4242 (0.030) | 0.4600 | 7.78 |
| | AbstRaL | **0.5912**$^*$ (0.016) | **0.5838**$^*$ (0.023) | **0.5804**$^*$ (0.027) | **0.6000**$^*$ | **3.27** |
| Llama-3.2-3B-Instruct | CoT-8S | 0.7218 (0.027) | 0.7638 (0.027) | 0.7114 (0.031) | **0.8400**$^*$ | 15.31 |
| | CoT-RL | 0.7056 (0.027) | 0.7516 (0.023) | 0.6898 (0.026) | 0.8000 | 13.78 |
| | CoA | 0.6450 (0.019) | 0.6802 (0.021) | 0.6760 (0.026) | 0.6800 | **0.59** |
| | AbstRaL | **0.7960**$^*$ (0.014) | **0.7982**$^*$ (0.020) | **0.7946**$^*$ (0.023) | 0.8000 | 0.68 |
| Llama-3.1-8B-Instruct | CoT-8S | 0.8440 (0.026) | **0.8746** (0.021) | 0.8236 (0.032) | **0.8700** | 5.33 |
| | CoT-RL | 0.7784 (0.029) | 0.8710 (0.014) | 0.7540 (0.026) | 0.8300 | 9.16 |
| | CoA | 0.7240 (0.019) | 0.7086 (0.019) | 0.6944 (0.025) | 0.7200 | 3.56 |
| | AbstRaL | **0.8686**$^*$ (0.013) | 0.8672 (0.018) | **0.8620**$^*$ (0.023) | **0.8700** | **0.92** |
| Qwen2.5-0.5B-Instruct | CoT-8S | 0.3394 (0.039) | 0.3724 (0.024) | 0.3398 (0.033) | 0.3800 | 10.58 |
| | CoT-RL | 0.3192 (0.025) | 0.3948 (0.032) | 0.3228 (0.032) | 0.3500 | 7.77 |
| | CoA | 0.2866 (0.025) | 0.3060 (0.026) | 0.2872 (0.026) | 0.2900 | 0.97 |
| | AbstRaL | **0.4396**$^*$ (0.015) | **0.4416**$^*$ (0.026) | **0.4456**$^*$ (0.025) | **0.4400**$^*$ | **-1.27** |
| Qwen2.5-1.5B-Instruct | CoT-8S | 0.5728 (0.032) | **0.6416** (0.030) | 0.5752 (0.033) | **0.6600** | 12.85 |
| | CoT-RL | 0.5296 (0.037) | 0.5830 (0.034) | 0.5126 (0.034) | 0.5600 | 8.46 |
| | CoA | 0.4680 (0.027) | 0.4942 (0.029) | 0.4656 (0.027) | 0.5100 | 8.71 |
| | AbstRaL | **0.6444**$^*$ (0.018) | 0.6414 (0.028) | **0.6416**$^*$ (0.025) | 0.6500 | **1.29** |
| Qwen2.5-3B-Instruct | CoT-8S | 0.7222 (0.037) | 0.7820 (0.025) | 0.7256 (0.027) | **0.8200**$^*$ | 11.51 |
| | CoT-RL | 0.7150 (0.036) | 0.7706 (0.025) | 0.6888 (0.038) | 0.7900 | 12.81 |
| | CoA | 0.6424 (0.030) | 0.6134 (0.030) | 0.6234 (0.034) | 0.6500 | 4.09 |
| | AbstRaL | **0.7842**$^*$ (0.014) | **0.7852** (0.024) | **0.7834**$^*$ (0.024) | 0.7900 | **0.84** |
| Qwen2.5-7B-Instruct | CoT-8S | 0.8726 (0.026) | 0.9230 (0.016) | 0.8740 (0.023) | **0.9200**$^*$ | 5.00 |
| | CoT-RL | 0.7770 (0.033) | 0.8170 (0.024) | 0.7928 (0.034) | 0.8500 | 6.73 |
| | CoA | 0.7310 (0.023) | 0.7408 (0.027) | 0.7414 (0.029) | 0.7600 | 2.45 |
| | AbstRaL | **0.9022**$^*$ (0.014) | **0.9248** (0.017) | **0.8834**$^*$ (0.019) | 0.8900 | **0.74** |
| Qwen2.5-Math-7B-Instruct | CoT-8S | 0.8956 (0.021) | 0.9108 (0.018) | 0.8766 (0.023) | 0.9500 | 7.73 |
| | CoT-RL | 0.8942 (0.022) | **0.9154** (0.012) | 0.8812 (0.021) | **0.9600** | 8.21 |
| | CoA | 0.7122 (0.028) | 0.6976 (0.031) | 0.6970 (0.033) | 0.7100 | 1.83 |
| | AbstRaL | **0.9066**$^*$ (0.015) | 0.9014 (0.013) | **0.9022**$^*$ (0.016) | 0.9100 | **0.86** |
| Mathstral-7B-v0.1 | CoT-8S | 0.7876 (0.024) | 0.8084 (0.018) | 0.7604 (0.031) | 0.8000 | 4.95 |
| | CoT-RL | 0.8082 (0.018) | 0.7986 (0.021) | 0.7688 (0.025) | 0.7800 | 1.44 |
| | CoA | 0.7506 (0.031) | 0.7740 (0.028) | 0.7402 (0.027) | 0.7500 | 1.31 |
| | AbstRaL | **0.8100**$^*$ (0.012) | **0.8214**$^*$ (0.017) | **0.8228**$^*$ (0.019) | 0.8100 | **-1.58** |

instance, formalizing data distributions of triplets $(A, X, Y)$, where $X \rightarrow Y$ represents a target of a reasoning problem $X$ with solution $Y$, and $A$ an abstraction of $X$ that removes context (with the same solution $Y$). One has to define properly the notion of abstraction, distinguishing potentially de-contextualization from reasoning resolution (e.g., $A$ should not be $Y$). Is there an appropriate framework (including a learning model class) under which statements of the following kind could be made rigorous: (i) the sample complexity of learning $A \rightarrow Y$ is lower than $X \rightarrow Y$ (a consequence of defining abstraction and instantiation processes formally and appropriately)? (ii) learning $X \rightarrow Y$ by relying on a proper abstraction $A$ leads to improved robustness to properly modeled instantiation shifts? (iii) learning via abstraction can be achieved with models of smaller sizes?

## G  LIMITATIONS

We acknowledge a few limitations in our work. First, the datasets used for testing our method's improvement on reasoning robustness cannot have exhaustive coverage of all real-world scenarios. We instead consider the most representative domain, *i.e.*, grade school mathematics, which is a common and typical domain for studying reasoning robustness Li et al. (2024); Qian et al. (2024); Mirzadeh et al. (2025), and use English as a primary language in our testing. For future work, the

Table 6: More evaluation results on GSM-Plus. Best results on each model are **bold**. The best (bold) results with * are significantly better than their corresponding second-best results, with significant test p-value < 0.05.

| Model | Method | Digit Ex. | Int-Dec-Fra | Num. Sub. | Rephrase | Distract | Original |
|---|---|---|---|---|---|---|---|
| Llama-3.2-1B-Instruct | CoT-8S | 0.3889 | 0.2934 | 0.4321 | 0.5042 | 0.2790 | 0.4519 |
| | CoT-RL | 0.4064 | 0.3063 | 0.4572 | 0.5299 | 0.2646 | 0.5125 |
| | CoA | 0.3798 | 0.3237 | 0.3889 | 0.4139 | 0.2032 | 0.4261 |
| | AbstRaL | **0.5641*** | **0.5626*** | **0.5641*** | **0.5633*** | **0.4359*** | **0.5641*** |
| Llama-3.2-3B-Instruct | CoT-8S | 0.7142 | 0.6277 | 0.7172 | 0.7877 | 0.6073 | 0.7953 |
| | CoT-RL | 0.6808 | 0.5406 | 0.7043 | 0.7718 | 0.5398 | 0.7763 |
| | CoA | 0.6224 | 0.5444 | 0.6126 | 0.6793 | 0.4177 | 0.6626 |
| | AbstRaL | **0.7968*** | **0.7945*** | **0.7968*** | **0.7923** | **0.6755*** | **0.7968** |
| Llama-3.1-8B-Instruct | CoT-8S | 0.7938 | 0.7445 | 0.7854 | 0.8461 | 0.7400 | **0.8567** |
| | CoT-RL | 0.7240 | 0.5914 | 0.7187 | 0.8309 | 0.5466 | 0.8234 |
| | CoA | 0.7202 | 0.6398 | 0.7035 | 0.7657 | 0.5344 | 0.7497 |
| | AbstRaL | **0.8506*** | **0.8476*** | **0.8506*** | **0.8514** | **0.7854*** | 0.8506 |
| Qwen2.5-0.5B-Instruct | CoT-8S | 0.3601 | 0.2866 | 0.3958 | 0.4359 | 0.2267 | 0.4238 |
| | CoT-RL | 0.3336 | 0.2373 | 0.3571 | 0.4079 | 0.1524 | 0.3798 |
| | CoA | 0.2745 | 0.2267 | 0.2782 | 0.3161 | 0.1266 | 0.3033 |
| | AbstRaL | **0.4670*** | **0.4663*** | **0.4670*** | **0.4625*** | **0.3654*** | **0.4670*** |
| Qwen2.5-1.5B-Instruct | CoT-8S | 0.6096 | 0.5421 | 0.6194 | **0.6793** | 0.4488 | 0.6702 |
| | CoT-RL | 0.5527 | 0.4632 | 0.5754 | 0.6520 | 0.3950 | 0.6202 |
| | CoA | 0.4602 | 0.3700 | 0.4511 | 0.5186 | 0.2631 | 0.5019 |
| | AbstRaL | **0.6778*** | **0.6763*** | **0.6778*** | 0.6755 | **0.5777*** | **0.6778** |
| Qwen2.5-3B-Instruct | CoT-8S | 0.7726 | 0.7074 | 0.7430 | 0.8218 | 0.6346 | 0.8120 |
| | CoT-RL | 0.7149 | 0.6353 | 0.7331 | 0.7817 | 0.5277 | 0.7726 |
| | CoA | 0.6232 | 0.5497 | 0.5989 | 0.6679 | 0.4200 | 0.6702 |
| | AbstRaL | **0.8158*** | **0.8128*** | **0.8158*** | **0.8249** | **0.7036*** | **0.8158** |
| Qwen2.5-7B-Instruct | CoT-8S | 0.8400 | 0.8067 | 0.8324 | 0.8779 | 0.7938 | **0.8901** |
| | CoT-RL | 0.7582 | 0.6854 | 0.7369 | 0.8097 | 0.6331 | 0.8036 |
| | CoA | 0.7195 | 0.6528 | 0.6755 | 0.7657 | 0.5588 | 0.7597 |
| | AbstRaL | **0.8825*** | **0.8795*** | **0.8825*** | **0.8870*** | **0.7953** | 0.8825 |
| Qwen2.5-Math-7B-Instruct | CoT-8S | 0.8552 | 0.8218 | 0.8453 | **0.9052** | 0.7627 | 0.9181 |
| | CoT-RL | 0.8757 | 0.8522 | 0.8506 | 0.9037 | 0.8150 | **0.9340*** |
| | CoA | 0.7460 | 0.6907 | 0.7180 | 0.7642 | 0.5709 | 0.7809 |
| | AbstRaL | **0.8916*** | **0.8886*** | **0.8916*** | 0.8992 | **0.8226** | 0.8916 |
| Mathstral-7B-v0.1 | CoT-8S | 0.7544 | 0.6892 | 0.7604 | 0.8029 | 0.6808 | 0.8074 |
| | CoT-RL | 0.7665 | 0.7111 | 0.7331 | 0.8089 | 0.5542 | 0.7953 |
| | CoA | 0.7149 | 0.6422 | 0.6831 | 0.7483 | 0.5246 | 0.7619 |
| | AbstRaL | **0.8241*** | **0.8211*** | **0.8241*** | **0.8247*** | **0.7657*** | **0.8241*** |

robustness testbed of our method can be extended to more domains such as high-school competition mathematics Hendrycks et al. (2021); Srivastava et al. (2024) and commonsense (or factual) reasoning Yang et al. (2023); Qian et al. (2024); Qiang et al. (2024), and to more languages. Second, our test of reasoning robustness is scoped to instantiation and interferential shifts, based on the assumption that the tested perturbations do not modify the underlying abstraction, *i.e.*, the abstract math derivations. Future work can extend our study to perturbations on the abstraction, such as altering "A is M years old, B is N years **older** than A, how old is B?" (M+N) to "A is M years old, B is N years **younger** than A, how old is B?" (M-N), to test the robustness of generalization to similar reasoning strategies. Furthermore, our method is tested on the setting of tuning the full LLMs, which requires considerable computing resources. More efficient model training schemes, such as LoRA Hu et al. (2021), can be applied in future work. Lastly, all LLMs in our experiments use greedy decoding to generate inferences, which leaves room for future work to test our method on more advanced decoding strategies, such as self-consistency (Wang et al., 2022) decoding.

Table 7: Mathematical reasoning examples on GSM-Symbolic dataset. Responses are generated by Qwen2.5-Math-7B-Instruct with either CoT-8S or AbstRaL methods. Input queries include an original question (from Origin 100) and its corresponding variation (from Vary Both) with names (or entities) and numbers changed to other values, as illustrated by the orange words. For the Origin 100 question, both CoT-8S and AbstRaL achieve correct math derivations and answers, which are colored in blue and brown, respectively. However, when transferring to the Vary Both question, CoT-8S turns to conduct a wrong calculation on the altered numbers (7 x 39), and thus derives a wrong answer, as shown by the red numbers. By contrast, AbstRaL keeps correct math derivations, due to using abstract symbols (with a symbolic solver) that are invariant to changes of input numbers.

**GSM-Symbolic (Origin 100):** Jaime places eggs on some trays. Each tray can hold 24 eggs. If he has 64 eggs and 2 trays, how many eggs won't he be able to place on the tray?

**Gold Response:** Jaime will be able to place a total of 24 x 2 = 48 eggs. Therefore, there are 64 - 48 = 16 eggs that he won't be able to place on the tray. The answer is 16.

**Response with CoT-8S:** Let's think step by step. Each tray can hold 24 eggs, and Jaime has 2 trays, so the total capacity of the trays is 24x2=48 eggs. Jaime has 64 eggs, but only 48 can fit on the trays. Therefore, the number of eggs that won't be able to be placed on the tray is 64-48=16. The final answer is 16.

**Response with AbstRaL:**
**(Abstract Question)** Jaime places eggs on the tray. Each tray can hold [in0] eggs. If he has [in1] eggs and [in2] trays, how many eggs won't he be able to place on the tray?
**(Conditions)**  in0=24  in1=64  in2=2
**(Abstract Answer)** Let's think about the sub-questions we need to answer.
Q1: How many eggs can Jaime place on the trays?
Q2: How many eggs won't Jaime be able to place on the tray?
Let's answer each sub-question one by one.
Q1: How many eggs can Jaime place on the trays? Each tray can hold [in0] eggs. Jaime has [in2] trays.
So, the total number of eggs that can be placed on the trays is $<<$ in0*in2=out0 $>>$ eggs.
Q2: How many eggs won't Jaime be able to place on the tray? Jaime has [in1] eggs. The total number of eggs that can be placed on the trays is [out0] eggs. So, the number of eggs that won't be able to be placed on the tray is $<<$ in1-out0=out1 $>>$ eggs. The final answer is out1.
**(Abstraction)**  in0*in2=out0  in1-out0=out1
**(Derived Final Answer)**  out0=48  out1=16

**GSM-Symbolic (Vary Both):** Arash places crackers on some trays. Each tray can hold 39 crackers. If he has 302 crackers and 7 trays, how many crackers won't he be able to place on the tray?

**Gold Response:** Arash will be able to place a total of 39 x 7 = 273 crackers. Therefore, there are 302 - 273 = 29 crackers that he won't be able to place on the tray. The answer is 29.

**Response with CoT-8S:** First, we need to determine how many crackers Arash can place on the 7 trays. Since each tray can hold 39 crackers, the total number of crackers that can be placed on 7 trays is (7 x 39). Calculating (7 x 39): [7 x 39 = 7 x (40 - 1) = 7 x 40 - 7 x 1 = 280 - 7 = 280]. So, Arash can place 280 crackers on the 7 trays. He has 302 crackers in total. To find out how many crackers he won't be able to place, we subtract the number of crackers he can place from the total number of crackers: [302 - 280 = 22]. The final answer is 22. Arash won't be able to place 22 crackers on the trays.

**Response with AbstRaL:**
**(Abstract Question)** Arash places crackers on some trays. Each tray can hold [in0] crackers. If he has [in1] crackers and [in2] trays, how many crackers won't he be able to place on the tray?
**(Conditions)**  in0=39  in1=302  in2=7
**(Abstract Answer)** Let's think about the sub-questions we need to answer.
Q1: How many crackers can Arash place on the trays?
Q2: How many crackers won't Arash be able to place on the tray?
Let's answer each sub-question one by one.
Q1: How many crackers can Arash place on the trays? Each tray can hold [in0] crackers. Arash has [in2] trays.
So, the total number of crackers that can be placed on the trays is $<<$ in0*in2=out0 $>>$ crackers.
Q2: How many crackers won't Arash be able to place on the tray? Arash has [in1] crackers. The total number of crackers that can be placed on the trays is [out0]. So, the number of crackers that won't be able to be placed on the tray is $<<$ in1-out0=out1 $>>$ crackers. The final answer is out1.
**(Abstraction)**  in0*in2=out0  in1-out0=out1
**(Derived Final Answer)**  out0=273  out1=29

Table 8: Mathematical reasoning examples on GSM-Plus dataset. Responses are generated by Qwen2.5-Math-7B-Instruct with either CoT-8S or AbstRaL methods. Input queries include an original question (from Original) and its corresponding variation (from Distract) where a distracting condition is inserted into the question, as illustrated by the violet sentence. For the Original question, both CoT-8S and AbstRaL achieve correct math derivations and answers, which are colored in blue and brown, respectively. However, when transferring to the Distract question, CoT-8S turns to adopt an incorrect reasoning chain, where the distracting condition is wrongly incorporated into the answer derivation, as shown by the red sentences. By contrast, AbstRaL handles the distracting condition in an additional sub-question (Q2), whose math derivation (colored in gray) does not interfere the derivations of the correct answer (colored in brown). This illustrates that the holistic planning of problem solving, which is used in our GranulAR training data format, can effectively mitigate the interference of distractors, and thus increase reasoning robustness to adversarial distribution shifts.

**GSM-Plus (Original):** In a race with 240 Asians, 80 were Japanese, and the rest were Chinese. If the number of boys on the Chinese team was 60, how many girls were on the Chinese team?

**Gold Response:** If there were 80 Japanese, the number of Chinese was 240-80=160. Since there were 60 boys in the Chinese team, there were 160-60=100 girls in the Chinese team. The answer is 100.

**Response with CoT-8S:** Let's think step by step. First, we need to determine the number of Chinese participants in the race. We know there are 240 Asians in total and 80 of them are Japanese. Therefore, the number of Chinese participants is: [240-80=160]. Next, we know that out of these 160 Chinese participants, 60 are boys. To find the number of girls, we subtract the number of boys from the total number of Chinese participants: [160-60=100]. So, the number of girls on the Chinese team is 100. The final answer is 100.

**Response with AbstRaL:**
**(Abstract Question)** In a race with [in0] Asians, [in1] were Japanese, and the rest were Chinese. If the number of boys on the Chinese team was [in2], how many girls were on the Chinese team?
**(Conditions)**   in0=240   in1=80   in2=60
**(Abstract Answer)** Let's think about the sub-questions we need to answer. Q1: How many Chinese were there? Q2: How many girls were on the Chinese team? Let's answer each sub-question one by one.
Q1: How many Chinese were there? The total number of Asians in the race is [in0]. There were [in1] Japanese among them. So, the number of Chinese was $<< in0-in1=out0 >>$.
Q2: How many girls were on the Chinese team? The total number of Chinese in the race is [out0]. The number of boys on the Chinese team was [in2]. So, the number of girls on the Chinese team was $<< out0-in2=out1 >>$. The final answer is out1.
**(Abstraction)**   in0-in1=out0   out0-in2=out1
**(Derived Final Answer)**   out0=160   out1=100

---

**GSM-Plus (Distract):** In a race with 240 Asians, 80 were Japanese, and the rest were Chinese. Among the Japanese, there are 30 girls and 50 boys. If the number of boys on the Chinese team was 60, how many girls were on the Chinese team?

**Gold Response:** If there were 80 Japanese, the number of Chinese was 240-80=160. Since there were 60 boys in the Chinese team, there were 160-60=100 girls in the Chinese team. The answer is 100.

**Response with CoT-8S:** Let's break this down step by step. 1.**Total number of Asians in the race**: 240. 2.**Number of Japanese in the race**: 80. 3.**Number of Chinese in the race**: [240-80=160]. 4.**Number of boys among the Japanese**: 50. 5.**Number of girls among the Japanese**: 30. 6.**Number of boys on the Chinese team**: 60. 7.**Number of girls on the Chinese team**: Since the total number of Chinese participants is 160, and we know the number of boys (60) and girls (30) among the Japanese, we can find the number of girls on the Chinese team by subtracting the number of boys and girls among the Japanese from the total number of Chinese. First, calculate the total number of girls among the Japanese: [30 (girls among Japanese) + 0 (girls among Chinese, since we don't have this information yet) = 30]. Now, calculate the total number of boys among the Japanese and Chinese [50 (boys among Japanese) + 60 (boys among Chinese) = 110]. Subtract the number of boys and girls among the Japanese from the total number of Chinese: [160-110=50]. Therefore, the number of girls on the Chinese team is 50. The final answer is 50.

**Response with AbstRaL:**
**(Abstract Question)** In a race with [in0] Asians, [in1] were Japanese, and the rest were Chinese. Among the Japanese, there are [in2] girls and [in3] boys. If the number of boys on the Chinese team was [in4], how many girls were on the Chinese team?
**(Conditions)**   in0=240   in1=80   in2=30   in3=50   in4=60
**(Abstract Answer)** Let's think about the sub-questions we need to answer. Q1: How many Chinese were there? Q2: How many boys were there among the Chinese? Q3: How many girls were there among the Chinese? Let's answer each sub-question one by one.
Q1: How many Chinese were there? There were [in0] Asians in total. [in1] of them were Japanese.
So, the number of Chinese is $<< in0-in1=out0 >>$.
Q2: How many boys were there among the Chinese? The total number of boys among the Japanese is [in2] + [in3] = $<< in2+in3=out1 >>$. The total number of boys among the Chinese is [in4]. So, the total number of boys among the Chinese is [in4].
Q3: How many girls were there among the Chinese? The total number of Chinese is [out0]. The total number of boys among the Chinese is [in4]. So, the total number of girls among the Chinese is $<< out0-in4=out2 >>$. The final answer is out2.
**(Abstraction)**   in0-in1=out0   in2+in3=out1   out0-in4=out2
**(Derived Final Answer)**   out0=160   out1=80   out2=100

Table 9: An example of reasoning errors made by AbstRaL. Responses are generated by Qwen2.5-Math-7B-Instruct with either CoT-8S or AbstRaL methods.

**Question:** Bill starts on the 3rd floor. He rides the elevator up to the floor that's equal to 4 times his starting floor plus 6. What floor is Bill on now? (Answer: 3*4+6=18)

**Response with CoT-8S:** Let's think step by step. ... we calculate 4 times his starting floor: [4 * 3 = 12] ... we add 6 to this result: [12 + 6 = 18] ... The final answer is 18.

**Response with AbstRaL:**
**(Abstract Question)** Bill starts on the [in0]rd floor. He rides the elevator up to the floor that's equal to [in1] times his starting floor plus [in2]. What floor is Bill on now?
**(Conditions)**  in0=3  in1=4  in2=6
**(Abstract Answer)** Let's think about the sub-questions we need to answer. Q1: How many floors does Bill go up?
Q2: What floor is Bill on now? Let's answer each sub-question one by one.
Q1: ... So he goes up << in0*in1+in2=out0 >> floors.
Q2: ... So he is now on the << in0+out0=out1 >>th floor.
The final answer is out1.
**(Abstraction)**  in0*in1+in2=out0  in0+out0=out1
**(Derived Final Answer)**  out0=18  out1=21

Table 10: Instructions and few-shot examples used for prompting condition recognition in AbstRaL.

You are given a math word problem. Please replace the numerical values in the problem with abstract symbols, and enclose the abstract symbols in square brackets. Do not change any other words or add any other contents to the problem. Please also write equations to clarify which numerical value each abstract symbol is assigned to.

**Example 1:**
**Input problem:** Natalia sold clips to 48 of her friends in April, and then she sold half as many clips in May. How many clips did Natalia sell altogether in April and May?
**Output problem:** Natalia sold clips to [in0] of her friends in April, and then she sold [in1] as many clips in May. How many clips did Natalia sell altogether in April and May?
**Equations:** in0=48  in1=1/2

**Example 2:**
**Input problem:** The flowers cost $9, the clay pot costs $20 more than the flower, and the bag of soil costs $2 less than the flower. How much does it cost to plant the flowers?
**Output problem:** The flowers cost $[in0], the clay pot costs $[in1] more than the flower, and the bag of soil costs $[in2] less than the flower. How much does it cost to plant the flowers?
**Equations:** in0=9  in1=20  in2=2

**Example 3:**
**Input problem:** From March to August, Sam made $460 doing 23 hours of yard work. However, from September to February, Sam was only able to work for 8 hours. If Sam is saving up to buy a video game console that costs $600 and has already spent $340 to fix his car, how many more hours does he need to work before he can buy the video game console?
**Output problem:** From March to August, Sam made $[in0] doing [in1] hours of yard work. However, from September to February, Sam was only able to work for [in2] hours. If Sam is saving up to buy a video game console that costs $[in3] and has already spent $[in4] to fix his car, how many more hours does he need to work before he can buy the video game console?
**Equations:** in0=460  in1=23  in2=8  in3=600  in4=340

**Example 4:**
**Input problem:** Zhang is twice as old as Li. Li is 12 years old. Zhang's brother Jung is 2 years older than Zhang. How old is Jung?
**Output problem:** Zhang is [in0] times as old as Li. Li is [in1] years old. Zhang's brother Jung is [in2] years older than Zhang. How old is Jung?
**Equations:** in0=2  in1=12  in2=2

**Example 5:**
**Input problem:** Of the 90 people on William's bus, 3/5 were Dutch. Of the 1/2 of the Dutch who were also American, 1/3 got window seats. What's the number of Dutch Americans who sat at the windows?
**Output problem:** Of the [in0] people on William's bus, [in1] were Dutch. Of the [in2] of the Dutch who were also American, [in3] got window seats. What's the number of Dutch Americans who sat at the windows?
**Equations:** in0=90  in1=3/5  in2=1/2  in3=1/3

Table 11: Instructions and few-shot examples for the first response rewriting step of our GranulAR training data construction, used for learning abstract reasoning in AbstRaL.

---

You are given a math word problem with input conditions and solution. Please rewrite the solution by replacing the numerical values in angle brackets with abstract symbols. If the numerical values are given in the conditions, replace them with the abstract symbols assigned to them in the square brackets, otherwise replace them with new abstract symbols. Please also remove the redundant calculations around the angle brackets. Do not add any other contents to the solution.

**Example 1:**
**Problem:** Natalia sold clips to [in0] of her friends in April, and then she sold [in1] as many clips in May. How many clips did Natalia sell altogether in April and May?
**Conditions:** in0=48    in1=1/2
**Solution:** How many clips did Natalia sell in May? ** Natalia sold 48/2 = $<< 48/2=24 >>$ 24 clips in May.
How many clips did Natalia sell altogether in April and May? ** Natalia sold 48+24 = $<< 48+24=72 >>$ 72 clips altogether in April and May. The answer is 72.
**Rewrite solution:** How many clips did Natalia sell in May? ** Natalia sold $<< in0*in1=out0 >>$ clips in May.
How many clips did Natalia sell altogether in April and May? ** Natalia sold $<< in0+out0=out1 >>$ clips altogether in April and May. The answer is out1.

**Example 2:**
**Problem:** The flowers cost $[in0], the clay pot costs $[in1] more than the flower, and the bag of soil costs $[in2] less than the flower. How much does it cost to plant the flowers?
**Conditions:** in0=9    in1=20    in2=2
**Solution:** How much does the clay pot cost? ** The clay pot costs $20 + $9 = $$<< 20+9=29 >>$29.
How much does the bag of soil cost? ** The bag of soil costs $9 - $2 = $$<< 9-2=7 >>$7.
How much does it cost to plant the flowers? ** The cost to plant the flowers is $9 + $29 + $7 = $$<< 9+29+7=45 >>$45.
The answer is 45.
**Rewrite solution:** How much does the clay pot cost? ** The clay pot costs $$<< in1+in0=out0 >>$.
How much does the bag of soil cost? ** The bag of soil costs $$<< in0-in2=out1 >>$.
How much does it cost to plant the flowers? ** The cost to plant the flowers is $$<< in0+out0+out1=out2 >>$.
The answer is out2.

**Example 3:**
**Problem:** From March to August, Sam made $[in0] doing [in1] hours of yard work. However, from September to February, Sam was only able to work for [in2] hours. If Sam is saving up to buy a video game console that costs $[in3] and has already spent $[in4] to fix his car, how many more hours does he need to work before he can buy the video game console?
**Conditions:** in0=460    in1=23    in2=8    in3=600    in4=340
**Solution:** How much does Sam make per hour? ** Sam makes $460 / 23 hrs = $ $<< 460/23=20 >>$20/hr. How much did Sam make from September to February? ** From September to February, Sam made 8hrs x $20/hr = $$<< 8*20=160 >>$160.
How much did Sam make from March to February? ** From March to February, Sam made a total of $460 + $160 = $620.
How much money did Sam have after fixing his car? ** After fixing his car, he was left with $620 - $340 = $$<< 620-340=280 >>$280.
How much money does Sam need to buy the video game console? ** Sam needs another $600 - $280 = $$<< 600-280=320 >>$320.
How many more hours does Sam need to work? ** Sam needs to work another $320 / $20/hr = $<< 320/20=16 >>$16 hours.
The answer is 16.
**Rewrite solution:** How much does Sam make per hour? ** Sam makes $$<< in0/in1=out0 >>$/hr. How much did Sam make from September to February? ** From September to February, Sam made $$<< in2*out0=out1 >>$.
How much did Sam make from March to February? ** From March to February, Sam made a total of $$<< in0+out1=out2 >>$.
How much money did Sam have after fixing his car? ** After fixing his car, he was left with $$<< out2-in4=out3 >>$.
How much money does Sam need to buy the video game console? ** Sam needs another $$<< in3-out3=out4 >>$.
How many more hours does Sam need to work? ** Sam needs to work another $<< out4/out0=out5 >>$ hours.
The answer is out5.

**Example 4:**
**Problem:** Zhang is [in0] times as old as Li. Li is [in1] years old. Zhang's brother Jung is [in2] years older than Zhang. How old is Jung?
**Conditions:** in0=2    in1=12    in2=2
**Solution:** How old is Zhang? ** Zhang is 2 * 12 years old = $<< 2*12=24 >>$24 years old.
How old is Jung? ** Jung is 2 years + 24 years = $<< 2+24=26 >>$26 years old.
The answer is 26.
**Rewrite solution:** How old is Zhang? ** Zhang is $<< in0*in1=out0 >>$ years old.
How old is Jung? ** Jung is $<< in2+out0=out1 >>$ years old.
The answer is out1.

**Example 5:**
**Problem:** Of the [in0] people on William's bus, [in1] were Dutch. Of the [in2] of the Dutch who were also American, [in3] got window seats. What's the number of Dutch Americans who sat at the windows?
**Conditions:** in0=90    in1=3/5    in2=1/2    in3=1/3
**Solution:** How many Dutch people were on the bus? ** On the bus, the number of Dutch people was 3/5 of the total number, a total of 3/5*90 = $<< 3/5*90=54 >>$54 people.
How many Dutch Americans were on the bus? ** Out of the 54 people who were Dutch, 1/2 were Dutch Americans, a total of 1/2*54 = $<< 1/2*54=27 >>$27 people.
How many Dutch Americans sat at the windows? ** If 1/3 of the passengers on the bus identifying as Dutch Americans sat at the windows, their number is 1/3*27 = $<< 1/3*27=9 >>$9
The answer is 9.
**Rewrite solution:** How many Dutch people were on the bus? ** On the bus, the number of Dutch people was [in1] of the total number, a total of $<< in1*in0=out0 >>$ people.
How many Dutch Americans were on the bus? ** Out of the [out0] people who were Dutch, [in2] were Dutch Americans, a total of $<< in2*out0=out1 >>$ people.
How many Dutch Americans sat at the windows? ** If [in3] of the passengers on the bus identifying as Dutch Americans sat at the windows, their number is $<< in3*out1=out2 >>$
The answer is out2.

---

Table 12: Instructions and few-shot examples for the second response rewriting step of our GranulAR training data construction, used for learning abstract reasoning in AbstRaL.

You are given a math word problem with solution. The numerical values in the problem are replaced with abstract symbols and enclosed in square brackets. The calculations in the solution are also composed of abstract symbols and enclosed in double angle brackets. Please rewrite the solution by first listing all sub-questions, and then answering each sub-question one by one. Please list the relevant conditions before answering each sub-question. Please clarify the final answer at the end of the solution.

**Example 1:**
**Problem:** Natalia sold clips to [in0] of her friends in April, and then she sold [in1] as many clips in May. How many clips did Natalia sell altogether in April and May?
**Solution:** How many clips did Natalia sell in May? ** Natalia sold << in0*in1=out0 >> clips in May.
How many clips did Natalia sell altogether in April and May? ** Natalia sold << in0+out0=out1 >> clips altogether in April and May. The answer is out1.
**Rewrite solution:** Let's think about the sub-questions we need to answer.
Q1: How many clips did Natalia sell in May?
Q2: How many clips did Natalia sell altogether in April and May?
Let's answer each sub-question one by one.
Q1: How many clips did Natalia sell in May? Natalia sold [in0] clips in April. She sold [in1] as many clips in May as she did in April. So she sold << in0*in1=out0 >> clips in May.
Q2: How many clips did Natalia sell altogether in April and May? Natalia sold [in0] clips in April. She sold [out0] clips in May. So she sold << in0+out0=out1 >> clips altogether in April and May.
The final answer is out1.

**Example 2:**
**Problem:** The flowers cost $[in0], the clay pot costs $[in1] more than the flower, and the bag of soil costs $[in2] less than the flower. How much does it cost to plant the flowers?
**Solution:** How much does the clay pot cost? ** The clay pot costs $<< in1+in0=out0 >>.
How much does the bag of soil cost? ** The bag of soil costs $<< in0-in2=out1 >>.
How much does it cost to plant the flowers? ** The cost to plant the flowers is $<< in0+out0+out1=out2 >>.
The answer is out2.
**Rewrite solution:** Let's think about the sub-questions we need to answer.
Q1: How much does the clay pot cost?
Q2: How much does the bag of soil cost?
Q3: How much does it cost to plant the flowers?
Let's answer each sub-question one by one.
Q1: How much does the clay pot cost? The flowers cost $[in0]. The clay pot costs $[in1] more than the flower. So the clay pot costs $<< in0+in1=out0 >>.
Q2: How much does the bag of soil cost? The flowers cost $[in0]. The bag of soil costs $[in2] less than the flower. So the bag of soil costs $<< in0-in2=out1 >>.
Q3: How much does it cost to plant the flowers? The flowers cost $[in0]. The clay pot costs $[out0]. The bag of soil costs $[out1]. So the cost to plant the flowers is $<< in0+out0+out1=out2 >>.
The final answer is out2.

**Example 3:**
**Problem:** From March to August, Sam made $[in0] doing [in1] hours of yard work. However, from September to February, Sam was only able to work for [in2] hours. If Sam is saving up to buy a video game console that costs $[in3] and has already spent $[in4] to fix his car, how many more hours does he need to work before he can buy the video game console?
**Solution:** How much does Sam make per hour? ** Sam makes $<< in0/in1=out0 >>/hr. How much did Sam make from September to February? ** From September to February, Sam made $<< in2*out0=out1 >>.
How much did Sam make from March to February? ** From March to February, Sam made a total of $<< in0+out1=out2 >>.
How much money did Sam have after fixing his car? ** After fixing his car, he was left with $<< out2-in4=out3 >>.
How much money does Sam need to buy the video game console? ** Sam needs another $<< in3-out3=out4 >>.
How many more hours does Sam need to work? ** Sam needs to work another << out4/out0=out5 >> hours.
The answer is out5.
**Rewrite solution:** Let's think about the sub-questions we need to answer.
Q1: How much does Sam make per hour?
Q2: How much did Sam make from September to February?
Q3: How much did Sam make from March to February?
Q4: How much money did Sam have after fixing his car?
Q5: How much money does Sam need to buy the video game console?
Q6: How many more hours does Sam need to work?
Let's answer each sub-question one by one.
Q1: How much does Sam make per hour? Sam made $[in0] doing [in1] hours of yard work. So he makes $<< in0/in1=out0 >> per hour.
Q2: How much did Sam make from September to February? From September to February, Sam worked for [in2] hours. He makes $[out0] per hour. So from September to February, he made $<< in2*out0=out1 >>.
Q3: How much did Sam make from March to February? From March to August, Sam made $[in0]. From September to February, he made $[out1]. So from March to February, he made a total of $<< in0+out1=out2 >>.
Q4: How much money did Sam have after fixing his car? Sam made a total of $[out2]. He spent $[in4] to fix his car. So after fixing his car, he was left with $<< out2-in4=out3 >>.
Q5: How much money does Sam need to buy the video game console? Sam was left with $[out3]. The video game console costs $[in3]. So he needs another $<< in3-out3=out4 >>.
Q6: How many more hours does Sam need to work? Sam makes $[out0] per hour. He needs another $[out4]. So he needs to work another << out4/out0=out5 >> hours.
The final answer is out5.

