# OpenReview forum: "AbstRaL: Augmenting LLMs' Reasoning by Reinforcing Abstract Thinking"
_ICLR.cc/2026/Conference — ICLR 2026 Poster_

### Official Review · Reviewer_vEZ7 · 2025-10-19

**Soundness:** 3
**Presentation:** 3
**Contribution:** 3
**Rating:** 8
**Confidence:** 4

**Summary:**

This paper introduces a novel reasoning technique that abstracts mathematical problems *before* solving them. The abstraction process goes from recognizing input variables to generating an abstraction that can be solved using external math tools or by an LLM tailored for the task. The authors show that this strategy improves the performance of many LLMs on two variants of the GSM dataset.

**Strengths:**

- The idea of uncovering the underlying abstraction of a math problem and using this abstraction to solve it (rather than the potentially noisy original text) is elegant.
- The abstraction pipeline can be implemented with relatively simple components, which means this direction is likely to used in practice.
- The results are consistently better than multiple strong baselines, for nine LLMs and two variants of the GSM dataset.

**Weaknesses:**

- While I think the contributions of this paper are sufficient for a conference publication, the authors focus on a single style of math problems since both datasets are spinoffs of the GSM dataset. The paper's claims would be stronger if at least another math dataset was used.
- In the same vein, previous work has consistently shown that mathematical reasoning problems are sensitive to noise in the problem formulation. How sensitive is this method to noise? See, for example, this new dataset for this investigation: https://ymingl.com/GSMDC/
- While I understand that math reasoning is the focus here, which, again, is sufficient for a conference paper, the authors should discuss how this work would be adapted for other types of reasoning scenarios.

**Questions:**

- What is the accuracy of each step of abstraction process summarized in Section 3?
- Eq 1: a better name for the reward formula introduced here would be "symbolic similarity reward" since it's computed as (1 - distance).
- Are the results shown in Table 1 statistically significantly better than the next strongest baseline in Table 1?
- Typo in Abstract: "we instead focuses"

---

> ### Author Response · Authors · 2025-11-21
> **Author Response to Reviewer vEZ7**
>
> We thank the reviewer for acknowledging that our idea of abstraction is elegant, and for approbating the practical value of our abstraction method and its consistent improvements upon multiple strong baselines on nine seed LLMs and two variants of the GSM dataset.
>
> \
> **1. (Weaknesses 1 and 3)** The reviewer concerns how our AbstRaL method would be applied to other math datasets beyond GSM and other types of reasoning scenarios.
>
> We first stress that current literature proposing robustness deficiencies in LLM reasoning is mostly concentrated in the GSM domain. Our paper therefore mainly aims to improve this **most prominent** GSM reasoning robustness. We concede that extending AbstRaL beyond GSM is an important extension, requiring us to first build corresponding robustness testing benchmarks, a whole project on its own, and then to re-formulate the more general abstraction process. For example, in other domains such as commonsense reasoning, input conditions could be entities or key phrases (each similarly assigned to an abstract symbol) mentioned in the question, and the abstraction could be a sequence of head-action-tail triples, where head and tail are abstract symbols representing input and output entities, and action represents a domain-specific operation or API call (e.g., search of relevant or linked entities in a knowledge graph or database). We leave the testing of AbstRaL on other variation datasets for future work, which is out of the scope of this paper.
>
> More importantly, in Section 5.2, we have justified that AbstRaL already implicitly benefits LLM reasoning on a wide range of **OOD datasets**. This shows that, despite the abstraction data being developed for a specific domain, the abstraction process does propagate to some extent beyond that specific domain.
>
>
> \
> **2. (Weakness 2)** The reviewer asks how our AbstRaL method is sensitive to noise added to the reasoning problem formulation, and suggests investigation on related datasets such as GSM-DC.
>
> We clarify that LLMs trained with AbstRaL achieve better robustness to irrelevant contexts or distractors, i.e., are **less sensitive to noise** in the problem. Similar to GSM-DC, we have tested our method on the Distract perturbation in GSM-Plus dataset (in Table 1), which adds topic-related but useless conditions as noisy contexts or distractors into the reasoning problem. Compared to baseline methods, AbstRaL more effectively mitigates the interference of such distractive noise.
>
> \
> **3. (Question 1)** The reviewer asks for stratified accuracy of each abstraction step (i.e., condition recognition, abstract reasoning, abstraction retrieval and symbolic derivation) of our AbstRaL framework.
>
> We clarify that in our studied GSM domain, the first (condition recognition) step of AbstRaL is a task nearly solved by the neural parser (i.e., the oracle LLM), and the third (abstraction retrieval) and last (symbolic derivation) steps of AbstRaL are both rule-based tasks that can be precisely executed by existing (Python) scripts or toolkits. Therefore, the accuracy of the above three steps is (almost or exactly) 100%, and the accuracy of the second (abstract reasoning) step, which is the core step of AbstRaL, is roughly identical to the final GSM problem answer accuracy shown in our experimental results.
>
> \
> **4. (Question 3)** The reviewer asks whether our (best) results shown in Table 1 are statistically significantly better than the next strongest baseline results.
>
> We clarify that in our full experimental results in Table 4 and 5 (in Appendix D), we have labeled our best results (with the star sign) if they are statistically significantly better than their corresponding second-best results (with significant test p-value < 0.05). The best results achieved by AbstRaL are mostly statistically significantly better than the next strongest baseline results.
>
> \
> **5. (Questions 2 and 4)** The reviewer suggests renaming our proposed “symbolic distance reward” as “symbolic similarity reward”, and fixing the typo “we instead focuses” in Abstract.
>
> We thank the reviewer for their recommended naming and identified typo, and will revise them in the camera-ready revision.

---

> > ### Comment · Reviewer_vEZ7 · 2025-11-21
> >
> > Thank you for your response!

---

> > > ### Author Response · Authors · 2025-11-27
> > >
> > > Dear Reviewer vEZ7,
> > >
> > > We truly appreciate your further comment, and want to make sure whether all your raised points are resolved, clarified or justified in our rebuttal?
> > >
> > > Thank you!

---

> > > > ### Comment · Reviewer_vEZ7 · 2025-11-28
> > > >
> > > > Yes, you have resolved/explained the points I raised. Thank you!
> > > > I think the overall grade of 8 is fair for the contribution of this work.

---

> > > > > ### Author Response · Authors · 2025-11-28
> > > > >
> > > > > Dear Reviewer vEZ7,
> > > > >
> > > > > Thank you for your confirmation and positive rating!

---

### Official Review · Reviewer_XBfh · 2025-11-01

**Soundness:** 3
**Presentation:** 3
**Contribution:** 4
**Rating:** 8
**Confidence:** 4

**Summary:**

This paper presents an interesting technique of leveraging symbolic abstract thinking as a way to improve the performance of LLM's ability to solve math problems. Instead of directly direct solutions from surface form, LLMs recognize input parameters, symbolic formula to compute the answer, and then the answer is derived by symbolic execution.

The paper presents a data generation pipeline, that (1) rewrites instance into parametric representation, (2) sample CoT, and (3) rewrite CoT into abstract CoT and use symbolic derivation to calculate final answers. This way, the model can be trained with first SFT and then RL to perform abstract reasoning.

Results show the performance of Abstral model on par and better than other reasoning based approaches. Ablation shows the effectiveness of RL, contexts and the reward design.

**Strengths:**

- This paper presents a quite interesting approach for math reasoning. Instead of directly reason on the text space, it leverages formula inference to derive the right result. This effectively turn math reasoning into a programming task (i.e., extract parameter, write a program, and run the program to get result) and assemble results with response.

- The dataset construction leverages existing CoT to construct generalizable abstract trace.

- Evaluation is pretty solid, especially with ablation with SFT vs RL's contribution to the final performance.

**Weaknesses:**

- the proposed abstraction composition is quite complicated, especially given the rewriting + retrieval (I personally think it's better called extraction as I was confused with document retrieval at first glance). Will a simpler way to construct reasoning, i.e, ask the model to directly generate formulas on top of basic COT (e.g., in a separate section) achieve the similar performance? Especially considering directly generate a small snippet of code for computation and ask LLM to assemble back to response. Is this just a format / style issue, or some more fundamental difference?

- r_symbolic seems quite important. But it's not clear reasons behind difference of results without no r_symbolic, some qualitative analysis compare results with and without r_symbolic would be helpful. I think r_symbolic is important kinda hint there may not be sufficient SFT for guiding formula generation.

**Questions:**

check the section above.

---

> ### Author Response · Authors · 2025-11-21
> **Author Response to Reviewer XBfh**
>
> We thank the reviewer for recognizing that our proposed method is interesting and effective for improving math reasoning, and for acknowledging our generalizable abstraction data construction pipeline and our solid experimental study.
>
> \
> **1. (Weakness 1)** The reviewer concerns that our proposed GranulAR abstraction schema is complicated, and asks whether a simpler reasoning schema (e.g., directly generating formulas on top of CoT, or code snippets) would achieve similar performance or have more fundamental difference.
>
> We clarify that the schema (or format) of abstraction **fundamentally affects the resulting robustness** of learned abstract reasoning, and a simpler reasoning schema is unlikely to achieve similar performance. This can be first verified by our ablation study (in Table 2) where AbstRaL - w/o GranulAR falls behind unablated AbstRaL on reasoning robustness, and AbstRaL - w/o Contexts, which directly generates formulas, also suffers from significant performance drop. Furthermore, we have tested that adding CoT reasoning before the direct generation of formulas does **not effectively improve** the performance, which indicates that interleaved abstraction and CoT is important for learning abstract reasoning (though more complicated). Besides, in terms of directly generating code snippets as abstraction, we have tested two related baseline methods (PoT and SyReLM in Table 1) that formulate abstraction based on programming or formal languages. Our results show that code-based abstract reasoning schema under-performs our proposed GranulAR schema based on the natural language.
>
> \
> **2. (Weakness 2)** The reviewer asks for more qualitative analysis to compare results with and without using our symbolic distance reward (r_symbolic) in RL.
>
> We claim that our symbolic distance reward more closely monitors the faithfulness of intermediate abstract reasoning steps, by checking the fine-grained abstraction similarity to the gold reference, which can **avoid learning erroneous reasoning shortcuts** that coincidentally got the correct answer. For example, in a case study, we find that the model trained without r_symbolic generates a wrong reasoning step *in0+in1=out0*, while the model trained with r_symbolic generates the correct one *in0\*in1=out0*. The original input conditions are *in0=2* and *in1=2*, so both formulas could get the correct answer *out0=4* (and both could get the answer correctness reward if the case happens in RL training). However, when the input conditions are perturbed to *in0=3* and *in1=4*, the wrong formula generated by the model without r_symbolic will get a wrong answer, leading to a performance drop in the robustness testing.

---

> ### Author Response · Authors · 2025-11-27
>
> Dear Reviewer XBfh,
>
> We sincerely appreciate your further comment at current author–reviewer discussion period, and would be grateful if you could verify whether all your raised points are resolved, clarified or justified in our rebuttal. We understand this is a busy time, and we truly value the time and effort you've dedicated to reviewing our work and engaging with our responses.
>
> Thank you!

---

### Official Review · Reviewer_wvec · 2025-11-01

**Soundness:** 2
**Presentation:** 3
**Contribution:** 2
**Rating:** 2
**Confidence:** 4

**Summary:**

This paper introduces AbstRaL, a framework designed to enhance the reasoning robustness of LLMs, particularly for GSM problems. Instead of relying on extensive data augmentation ("instantiation"), the authors propose to explicitly teach LLMs to first generate an abstract representation of a problem. This abstraction, being invariant to surface-form variations, is intended to make the model more robust to OOD shifts like changes in numbers, names, or the addition of distractors. The core of the method involves a new, fine-grained data format called GranulAR and a RL scheme built on top of SFT. The RL component uses novel, model-free rewards based on final answer correctness and the symbolic similarity of the generated abstraction to a gold reference, aiming to improve the faithfulness of the abstract reasoning process. The authors demonstrate that AbstRaL significantly improves performance on GSM robustness benchmarks (GSM-Symbolic, GSM-Plus) and shows some generalization benefits to other OOD tasks.

**Strengths:**

1. The central idea of moving from "instantiation" to "abstraction" is a strong conceptual contribution. Explicitly teaching the model a reasoning schema that is invariant to surface details is a promising direction for improving OOD robustness.

2.  The proposed AbstRaL framework is well-thought-out. The creation of the GranulAR data format, which blends socratic decomposition with symbolic abstraction, is a clever way to make the task more tractable for LLMs. The use of reinforcement learning with a custom symbolic distance reward (`rsymbolic`) to enforce faithfulness is a significant and novel part of the method.

**Weaknesses:**

1.  The framework's dependence on a vastly more powerful model (Llama-3.3-70B) for critical data generation steps is a major weakness. Both the initial "Condition Recognition" and the subsequent "Abstract Reasoning Chain Rewriting" are performed by this moel. This raises the concern that the improvements seen in smaller models are largely due to knowledge distillation from a superior model, rather than the inherent power of the AbstRaL learning scheme itself.

2. The method is evaluated exclusively on grade school math problems, where the underlying abstractions (arithmetic operations) are simple and well-defined. It is highly uncertain how this framework would translate to more complex domains with ambiguous or non-mathematical abstractions, such as legal reasoning, commonsense planning, or scientific hypothesis generation. The paper does not discuss these limitations or provide a path toward generalization.

3.   The primary comparison is between AbstRaL (SFT + RL) and baselines that are either prompting-based (CoT-8S) or SFT-only (CoA, AoT). The `w/o RL` ablation in Table 2 shows a massive performance drop, suggesting that the GranulAR data format with SFT alone is not very effective. A more direct and fair comparison would be between "AbstRaL-SFT" and other SFT-based abstraction methods. Furthermore, for the stronger Qwen2.5-Math-7B model, AbstRaL underperforms CoT-RL on the original, in-distribution data (91.0 vs 96.0), indicating a potential trade-off where improving OOD robustness comes at the cost of ID performance.

4.  The paper claims that AbstRaL "implicitly benefit more general reasoning capabilities." However, the results in Table 3 show that the improvements on general reasoning benchmarks like MMLU are marginal at best. The most significant gains are on other math-related datasets (MATH, AIME24), which is less surprising and does not fully support the claim of broad generalizability.

**Questions:**

1.  How can you be sure that the performance gains are not primarily a result of distilling its superior reasoning capabilities into the smaller models? Have you tried using a weaker model as the oracle to see how that affects the final performance?

2.  The `w/o RL` ablation shows that SFT on GranulAR data performs poorly, often worse than simple CoT prompting. Does this imply that the proposed abstraction format is too difficult for models to learn via SFT alone and that the complex RL setup is a necessary crutch?

3.  Regarding the scalability to other domains: What do you foresee as the main challenges in applying AbstRaL to problems where the "abstraction" is not a simple mathematical formula? For example, in a commonsense reasoning task, what would the "conditions" C and the "abstraction" A look like, and how would they be reliably extracted?

4.  For the strongest models, AbstRaL seems to trade in-distribution performance for OOD robustness (e.g., Table 1, Qwen2.5-Math-7B, Origin 100). Do you consider this an inherent trade-off of the method? Why do you think this performance degradation on the original task occurs?

---

> ### Author Response · Authors · 2025-11-21
> **Author Response (Part 1) to Reviewer wvec**
>
> We thank the reviewer for approbating our conceptual contribution and promising research direction towards improving OOD robustness, and for recognizing that our proposed AbstRaL framework is of good novelty and well-thought-out.
>
> \
> **1. (Weakness 1 and Question 1)** The reviewer concerns whether the improvements of our AbstRaL framework are largely due to a distillation of superior reasoning capabilities from the oracle LLM (Llama-3.3-70B-Instruct, which is used for input condition recognition and constructing our GranulAR reasoning traces), and whether a weaker LLM can be used as the oracle instead.
>
> The short answer is that AbstRaL does not necessarily rely on a strong oracle LLM, and a weaker LLM can be used as the oracle instead.
>
> We clarify that both input condition recognition and GranulAR data construction in AbstRaL are based on re-writing the existing Socratic CoT data. The oracle LLM only serves to **re-write the surface-form format** of existing reasoning traces, i.e., replacing real numerical values with abstract symbols and re-arranging the decomposed sub-question answering. Therefore, the improvements of our framework are **not due to distilling** more superior reasoning knowledge from the oracle LLM, but due to the AbstRaL learning scheme itself.
>
> We have verified in our ablation study (Table 2 - w/o Tools) that replacing the oracle LLM with the reasoning LLM itself to conduct input condition recognition results in only minor performance drop.
>
> We have further tested to use a **weaker** LLM (Llama-3.1-8B-Instruct) to re-write our GranulAR reasoning traces and conduct AbstRaL training based on that, which achieves **roughly the same** scores on our tested GSM perturbation and OOD datasets. These justify that it’s mainly the better learning of abstract reasoning that contributes to the improvements of AbstRaL. We will include these discussions and results in the camera-ready revision.
>
> \
> **2. (Weakness 2 and Question 3)** The reviewer asks how our AbstRaL method would be applied to more complex domains with ambiguous or non-mathematical abstractions.
>
> We first stress that current literature proposing robustness deficiencies in LLM reasoning is mostly concentrated in the GSM domain. Our paper therefore mainly aims to improve this **most prominent** GSM reasoning robustness. We concede that extending AbstRaL beyond GSM is an important direction, requiring us to re-formulate more general abstractions and to build corresponding robustness testing benchmarks. For example, in more complex domains such as commonsense reasoning, input conditions could be entities or key phrases (each similarly assigned to an abstract symbol) mentioned in the question, and the abstraction could be a sequence of head-action-tail triples, where head and tail are abstract symbols representing input and output entities, and action represents a domain-specific operation or API call (e.g., search of relevant or linked entities in a knowledge graph or database). We leave the domain extension of AbstRaL for future work, which is out of the scope of this paper.
>
> More importantly, in Section 5.2, we have justified that AbstRaL already implicitly benefits LLM reasoning on a wide range of **OOD datasets**. This shows that, despite the abstraction data being developed for a specific domain, the abstraction process does propagate to some extent beyond that specific domain.
>
> \
> **3. (Weakness 3)** The reviewer states that our experimental comparisons are primarily between AbstRaL (SFT+RL) and baselines that are either prompting-based (CoT-8S) or SFT-only (CoA, AoT), and asks for a comparison between SFT-only AbstRaL (without RL) and the SFT-based abstraction methods.
>
> We clarify that the statement of our experimental comparisons is not true. We also compare our AbstRaL method to CoT-RL that applies the same RL (without the abstraction), and SyReLM that similarly applies a warm-up SFT and a RL with symbolic rewards. We have introduced all baseline methods in Appendix C.
>
> SFT-only AbstRaL still outperforms SFT-based baseline methods due to a better formulation of abstract reasoning traces (GranulAR). We clarify that AbstRaL - w/o RL (in Table 2) scores consistently higher than CoA and AoT (in Table 1).

---

> ### Author Response · Authors · 2025-11-21
> **Author Response (Part 2) to Reviewer wvec**
>
> \
> **4. (Weakness 3 and Question 4)** The reviewer points out a potential trade-off: on some strong seed LLMs such as Qwen2.5-Math-7B-Instruct, our AbstRaL method underperforms baseline CoT methods on the original (unperturbed) GSM8K testset. The reviewer also concerns why this performance degradation on the original task occurs.
>
> We acknowledge that on strong seed LLMs that already achieve high accuracies on the original GSM8K, AbstRaL may trade off minor scores on these original testing samples for better reasoning robustness against perturbations and higher reasoning accuracies on OOD tasks. We view such **trade off as a strength** of our method, which mitigates LLMs’ overfitting to the existing in-distribution testing samples.
>
> **Error Analysis**:
> AbstRaL teaches LLMs to analyze and decompose reasoning problems at the start of inference, which requires LLMs to perform more holistic problem understanding and solution planning, compared to CoT. This however may lead to slightly overthinking and thus misunderstanding of a few reasoning problems. We have manually checked the GSM8K testing questions where Qwen2.5-Math-7B-Instruct answered correctly with 8-shot CoT prompting (CoT-8S) but wrongly using AbstRaL (**totally 35** such questions), and verified that AbstRaL’s failure on all these questions are due to minor misunderstanding. For example, in the case below, Qwen2.5-Math-7B-Instruct (AbstRaL) misunderstood "go up to" as "go up by", and therefore did a redundant reasoning step. Our case study has justified that AbstRaL's score trade-off is due to such in-depth yet minor misunderstanding (rather than shallow misuse of unnecessary symbolic processing or patterns unassociated to the problem solving).
>
> Bill starts on the 3rd floor. He rides the elevator up to the floor that's equal to 4 times his starting floor plus 6. What floor is Bill on now? (Answer: 3*4+6=18)
>
> *Qwen2.5-Math-7B-Instruct (CoT-8S):*
>
> Let's think step by step. … we calculate 4 times his starting floor: [4 * 3 = 12] … we add 6 to this result: [12 + 6 = 18] … The final answer is 18.
>
> *Qwen2.5-Math-7B-Instruct (AbstRaL):*
>
> (Abstract Question) Bill starts on the [in0]rd floor. He rides the elevator up to the floor that's equal to [in1] times his starting floor plus [in2]. What floor is Bill on now?
>
> (Conditions) in0=3, in1=4, in2=6
>
> (Abstract Answer) Let's think about the sub-questions we need to answer. Q1: How many floors does Bill go up? Q2: What floor is Bill on now? Let's answer … Q1: … So he goes up <<in0*in1+in2=out0>> floors. Q2: … So he is now on the <<in0+out0=out1>>th floor. The final answer is out1.
>
> (Abstraction) in0*in1+in2=out0, in0+out0=out1
>
> (Derived Final Answer) out0=18, out1=21
>
> \
> **5. (Weakness 4)** The reviewer states that in Table 3, the improvements of AbstRaL on general reasoning benchmarks are marginal, which does not fully support our claim that AbstRaL broadly enables better generalizability.
>
> We clarify that as shown by our full experimental results in Table 7 (in Appendix D), AbstRaL achieves improvements **consistently on various** general reasoning benchmarks, including MMLU (all portions), BBH, ARC-Challenge and OpenBookQA. Although some improvements are marginal, others are notable and the consistent increase of scores on all datasets strongly supports our claim of broad generalizability.
>
> \
> **6. (Question 2)** The reviewer asks whether our proposed GranulAR abstraction format is too difficult for LLMs to learn via SFT alone, and thus necessarily requires complex RL.
>
> We first clarify that the format of abstraction significantly **affects the resulting robustness** of learned abstract reasoning. This can be verified by our ablation study (in Table 2) where AbstRaL - w/o GranulAR suffers from large robustness drop, and by the under-performance of baseline SyReLM (in Table 1) which uses a different abstraction format based on programming or formal languages. Therefore, learning with a proper abstraction format such as GranulAR is important (though difficult). We however concede that learning robust reasoning within an abstraction format is a **non-trivial** task, which also requires proper advanced learning approaches such as our RL with abstraction rewards.

---

> ### Author Response · Authors · 2025-11-27
>
> Dear Reviewer wvec,
>
> We sincerely appreciate your further comment at current author–reviewer discussion period, and would be grateful if you could verify whether all your raised points are resolved, clarified or justified in our rebuttal. We understand this is a busy time, and we truly value the time and effort you've dedicated to reviewing our work and engaging with our responses.
>
> Thank you!

---

### Official Review · Reviewer_AUyf · 2025-11-01

**Soundness:** 2
**Presentation:** 2
**Contribution:** 2
**Rating:** 2
**Confidence:** 4

**Summary:**

Authors introduce AbstRaL (Abstraction Reinforcement Learning), a framework that uses abstract intermediate representations to improving the robustness of LLMs in math word problems. Authors propose teaching models to abstract reasoning problems—that is, to represent problems symbolically and reason at a higher level of abstraction. AbstRaL integrates RL with a new Granularly-decomposed Abstract Reasoning (GranulAR) dataset that blends CoT reasoning with symbolic abstraction. The method encourages LLMs to generate faithful abstract representations that can be solved via symbolic tools such as equation solvers. Experiments are done two LLMs mainly and results on new benchmarks (GSM-Symbolic, GSM-Plus) show improved robustness under numerical, contextual, and distractor perturbations.

**Strengths:**

1. Combination of learning abstractions using an SFT-RL framework seems interesting.
2. Though authors have used only two main LLMs, the ablation studies are done htoughtfully.
3. Using the socratic version of GSM8k to rewrite the abstract intermediate steps (/decomposed questions) is interesting. Therefore, makes the data somewhat valuable.

**Weaknesses:**

The paper suffers largely on novelty, lack of sufficient experimentation, broad claims which are untrue.

Novelty: Neither decomposition nor abstraction is new in symbolic math, math word problem solving. There are too many papers, with too many variations and ideas. Authors also do not specify any contributions. So, I am not even sure if this is a central claim.

Lack of sufficient Experimentations: Only two small LMs are chosen to support very broad claims. Even the variations of GSM chosen are only two. There are many variants: GSM-Hard, i-GSM, GSM8k_MORE. And this is just GSM8k. There are so many fantastic and important datasets out there.

Broad claims: See below.

Others:
1. Limited comparison to earlier equation-prediction approaches: Prior work (e.g., Mitra et al. 2018; Agarwal et al. 2021) also translated math word problems into equations for symbolic solving (more below)
2. The generation of GranulAR data relies heavily on a large teacher model (Llama-3.3-70B). Whats the effect of using such a model here. What if we use smaller models.
3. While the method checks answer correctness, it is unclear how the necessary and sufficient quality of abstract decompositions is ensured or whether abstraction errors propagate through training.

**Questions:**

L77: There are other work which used to convert the math word problems to equations and then use tools to solve it. Why is this different?

For example, Agarwal et al. ICLR 2021 MathAI showed that it was fundamentally difficult for Transformers to perform numeric addition, multiplication. So, using symbols in place and postponing the calculation worked much better. Other work such as Mitra et al. 2018 and many, also predicted equations from math words as an intermediate step. This was the norm in pre-LLM era.

L201: The abstract answer $\mathcal{Y}_\mathcal{A}$ seems to be generated by LLMs. It involves a crucial step of decomposition which needs to have a necessary and sufficiency check? Was it done by any means?

L250: This will miss any equivalent expressions, lets say which is a simplification or complicated version of A.   Can't we look at AST matches?

L268: These types of claims and statements are very generic. Many LLMs are trained code corpora, sees a lot of different structured languages. Why should it not be inherently incapable of such abstraction?

L375: Doesn't it show that the step-by-step decomposition is helping more? Based on the drops in Table 2, thats what is implying. Am I missing anything here?

L401: Its interesting that removing the abstraction does not help much with distractors. But these are small models (only two). Would it affect larger ones as well? Is it possible to show this on large models (gpt-oss size).

---

> ### Author Response · Authors · 2025-11-21
> **Author Response (Part 1) to Reviewer AUyf**
>
> We thank the reviewer for finding our abstraction learning framework interesting, and for recognizing our thoughtful ablation analysis and valuable data collection.
>
> \
> **1. (Novelty, Other Weakness 1 and Question L77)** The reviewer asks for clarifying our contributions, and how our proposed AbstRaL method is novel and different from earlier approaches (e.g., Mitra et al., 2018; Agarwal et al., 2021) that convert math word problems to equations and then solve them by tools.
>
> We acknowledge that symbolic abstraction with decomposition from calculation is commonly studied in earlier works for math word problem solving. However, as verified in our ablation study (Section 5.1 and Table 2), improving reasoning robustness to distribution shift via such symbolic abstraction is not directly achieved by the LLMs. This requires more advanced learning approaches as studied in the paper, and our contribution uses  a **better data format and training scheme** to promote robust LLM reasoning via symbolic abstraction.
>
> Specifically, different from earlier approaches cited by the reviewer, AbstRaL proposes a **new reasoning data format** that integrates symbolic abstraction (or equation prediction) within chain-of-thought and socratic problem decomposition, and also a **new reinforcement learning scheme** with abstraction rewards. Both proposed abstraction techniques effectively improve the reasoning robustness of LLMs.
>
> Instead of testing the earlier approaches, we have compared AbstRaL to **more recent and advanced** symbolic abstraction approaches as baselines, including PoT (Chen et al., 2022), CoA (Gao et al., 2025), AoT (Hong et al., 2024) and SyReLM (Dutta et al., 2024). AbstRaL’s outperformances over these recent baselines have more convincingly verified our contributions to robust reasoning with abstraction.
>
> \
> **2. (Lack of Sufficient Experimentations)** The reviewer states that only two LLMs are tested in our experiments to support our claims.
>
> This is not true. We did include our full evaluation results seeded with more LLMs in Appendix D, covering three LLMs from Llama3 model series, five LLMs from Qwen2.5 model series, and Mathstral-7B. Comparisons of learning methods on these **9 LLMs** are consistent with comparisons on the two representative LLMs presented in the main body of our paper. We can move a summary back in the main body for the camera-ready revision.
>
> \
> **3. (Lack of Sufficient Experimentations)** The reviewer states that only two GSM variation datasets are tested in our experiments, and concerns about more GSM variation datasets (e.g., GSM-Hard, i-GSM and GSM8k_MORE) and datasets beyond GSM8K variations.
>
> We clarify that the two GSM variation datasets (i.e., GSM-Symbolic and GSM-Plus) chosen in our experiments are commonly studied benchmarks for testing GSM reasoning robustness, which include **diverse (totally 8) types** of instantiation shifts and interferential perturbations. Other GSM datasets suggested by the reviewer contain either similar variations that are **already covered** by our tested benchmarks, or variations that **modify the underlying abstraction** of the reasoning problem (e.g., adding a relevant condition to complicate the problem), which go beyond the scope of our study.
>
> We also stress that current literature proposing robustness deficiencies in LLM reasoning is mostly concentrated in the GSM domain. Our paper therefore mainly aims to improve this **most prominent** GSM reasoning robustness. We concede that extending AbstRaL beyond GSM is an important extension, requiring us to first build corresponding robustness testing benchmarks, a whole project on its own, and then to re-formulate the more general abstraction process. For example, in other domains such as commonsense reasoning, input conditions could be entities or key phrases (each similarly assigned to an abstract symbol) mentioned in the question, and the abstraction could be a sequence of head-action-tail triples, where head and tail are abstract symbols representing input and output entities, and action represents a domain-specific operation or API call (e.g., search of relevant or linked entities in a knowledge graph or database). We leave the testing of AbstRaL on other variation datasets for future work, which is out of the scope of this paper.
>
> More importantly, in Section 5.2, we have justified that AbstRaL already implicitly benefits LLM reasoning on a wide range of **OOD datasets**. This shows that, despite the abstraction data being developed for a specific domain, the abstraction process does propagate to some extent beyond that specific domain.

---

> ### Author Response · Authors · 2025-11-21
> **Author Response (Part 2) to Reviewer AUyf**
>
> \
> **4. (Other Weakness 2)** The reviewer concerns whether the improvements of our AbstRaL framework heavily relies on using a large oracle LLM (Llama-3.3-70B-Instruct) to construct our GranulAR training data, and whether a smaller LLM can be used instead.
>
> The short answer is that AbstRaL does not necessarily rely on a large oracle LLM, and a smaller LLM can be used instead.
>
> Our GranulAR training data is constructed based on re-writing the existing Socratic CoT data. So the large oracle LLM only serves to **re-write the surface-form format** of existing reasoning traces, i.e., re-arranging the decomposed sub-question answering, and replacing real numerical values with abstract symbols. Therefore, the improvements of our framework **do not rely on distilling** more superior reasoning knowledge from the large oracle LLM.
>
> We have tested to use a **smaller** LLM (Llama-3.1-8B-Instruct) to construct our GranulAR data and conduct AbstRaL training based on that, which achieves **roughly the same** scores on our tested GSM variation and OOD evaluation datasets. We will include these discussions in the camera-ready revision.
>
> \
> **5. (Other Weakness 3 and Question L201)** The reviewer concerns how the necessary and sufficient quality of abstract problem decomposition is ensured in our GranulAR data and AbstRaL training.
>
> As clarified in point 4, our GranulAR training data is a re-writing of the existing Socratic CoT data. So each constructed GranulAR trace is supposed to use the **same problem decomposition scheme** as that used in the original Socratic CoT trace, while Socratic CoT is manually annotated and thus ensures the necessary and sufficient quality of decomposition. Based on that, our AbstRaL training framework uses a **symbolic distance** reward in the reinforcement learning, which closely guides LLMs to learn the necessary and sufficient abstract decomposition labeled in the re-written GranulAR data.
>
> \
> **6. (Question L250)** The reviewer concerns that the symbolic distance reward used in our RL approach does not consider mathematical equivalent expressions when matching between generated and gold abstractions, and suggests more complex matching such as the distance of abstract syntax tree (AST).
>
> We concede that our proposed symbolic distance reward does not encourage the exploration of equivalent expressions in RL rollouts. However, as a trade-off, such strict reward **closely guides** LLMs to learn proper decomposition and faithful reasoning abstraction from the well-qualified gold references. As verified in our ablation study (Section 5.1 and Table 2), our symbolic distance reward effectively improves GSM reasoning accuracy and robustness to distribution shifts. We leave the investigation of other symbolic matching rewards for future work.
>
> \
> **7. (Question L268)** The reviewer asks why LLMs, which are often trained on code and structured languages besides natural language corpora, still fall short of predicting de-contextualized abstractions.
>
> We acknowledge that most current LLMs are also trained on code corpora and structured languages. However, as verified in our ablation study (Section 5.1 and Table 2), it is still hard for LLMs to generate GSM abstractions without intermediate contexts. We assume that this is because most reasoning data samples in LLM pre-training and post-training corpora are still in natural language format (e.g., chain-of-thought traces), and thus LLMs might fall short of integrating multi-step reasoning capabilities within the abstract symbolic format, which may be **less frequently seen** in the reasoning training data samples. We will clarify related claims in the camera-ready revision.
>
> \
> **8. (Questions L375 and L401)** The reviewer asks whether the problem decomposition plays a more important role compared to the symbolic abstraction in our GranulAR reasoning schema, especially for mitigating the effect of distractors.
>
> We clarify that the problem decomposition in GranulAR may need to be **rooted in** the symbolic abstraction, in order to effectively mitigate the interference of some distractors, e.g., as shown in our case study in Table 9, re-thinking of relevant abstract input conditions in each decomposed sub-question answering may be essential for excising out the distractors and useless sub-questions.

---

> ### Author Response · Authors · 2025-11-21
> **Author Response (Part 3) to Reviewer AUyf**
>
> \
> **9. (Question L401)** The reviewer asks whether AbstRaL is also effective on larger-size LLMs.
>
> We clarify that AbstRaL aims to improve the reasoning robustness of mainly smaller-size LLMs, because smaller LLMs have **more prominent robustness deficiencies**, as verified in the literature of our tested GSM-Plus and GSM-Symbolic benchmarks. The focus of our paper was thus mostly on this regime of main interest.
>
> Improving smaller LLMs without scaling up model parameters also contributes to **higher computational efficiency**. The largest LLMs (7B or 8B) tested in our paper take averagely ~2 seconds to inference per GSM testing sample on 1 A100 (80GB) GPU. By contrast, oracle-level LLMs (70B or 72B), if finetuned on Socratic CoT data, take averagely ~6 seconds to inference per sample, and need 4 A100 (80GB) GPUs.

---

> ### Author Response · Authors · 2025-11-27
>
> Dear Reviewer AUyf,
>
> We sincerely appreciate your further comment at current author–reviewer discussion period, and would be grateful if you could verify whether all your raised points are resolved, clarified or justified in our rebuttal. We understand this is a busy time, and we truly value the time and effort you've dedicated to reviewing our work and engaging with our responses.
>
> Thank you!

---

> > ### Comment · Reviewer_AUyf · 2025-11-27
> > **About 9 LLMs**
> >
> > Noted the experiment about 9 LLMs. I will update my original review.
> >
> > I am, however, still not convinced about novelty. The rebuttal seems to be ripe with the same issues as the paper. How is equation prediction anywhere new? This is the first novelty most "math word" problem modelers tried to use as an intermediate representation.
> > The use of <in0> etc has been seen at many places. For example, SKiC prompting (skills-in-context) does that invoke tools using these abstractions as special token markers. This is just one. A simple search would lead to many papers with similar ideas. Provided the data construction with abstraction is one of the central points, I am unsure of its impact on top of the existing body of work.
> > Another claim revolves around abstraction learnability using RL. In general, most work [1] suggests RL is better than SFT in finetuning LLMs. Even if this is specific to abstraction learning, shouldn't we vary various types of abstract representations and then show how RL generalizes.
> >
> > [1] SFT Memorizes, RL Generalizes: A Comparative Study of Foundation Model Post-training

---

> > > ### Author Response · Authors · 2025-11-29
> > >
> > > Dear Reviewer AUyf,
> > >
> > > Thank you for your follow-up comment, we greatly appreciate it and provide more clarifications below.
> > >
> > > \
> > > **10. (Novelty of Abstraction and Data Construction)** The reviewer concerns that the idea of equation prediction (or abstraction) as an intermediate representation for math word problem solving is not new, and thus how our GranulAR data construction with abstraction contributes to the existing body of work.
> > >
> > > We first clarify that the novelty of our paper reflects in proposing a **new abstraction framework to improve LLMs’ reasoning robustness to distribution shifts**. Although we acknowledge that the idea of equation prediction (or abstraction) as an intermediate representation is not new, existing work on this track overlooks its potential to benefit the robustness aspect. Our work therefore innovatively promotes the study of **bridging abstraction and robustness** in LLM reasoning.
> > >
> > > In terms of the contributions related to our GranulAR data construction, we clarify that the **schema (or format) of abstraction significantly affects the resulting robustness** of learned abstract reasoning, and other abstraction schemas (previously proposed in existing work) are unlikely to achieve comparable performance. This can be first verified by our ablation study (in Table 2) where AbstRaL - w/o GranulAR, which uses a previously proposed (socratic CoT) format of abstraction, falls behind unablated AbstRaL on reasoning robustness, and AbstRaL - w/o Contexts, which directly generates formulas, also suffers from significant performance drop. Besides, we have tested two related baseline methods (PoT and SyReLM in Table 1) that formulate abstraction based on programming or formal languages, both under-performs our proposed GranulAR schema. Therefore, learning with a proper abstraction schema is essential to improve reasoning robustness, and our GranulAR data construction **contributes to an improved schema** of abstraction on top of the existing work.
> > >
> > > \
> > > **11. (RL on Various Abstractions)** The reviewer asks for testing how RL generalizes to various types of abstract representations.
> > >
> > > As clarified in our above response point 10, we have tested RL on various schemas of abstractions in our baseline comparisons (Table 1) and ablation study (Table 2), including SyReLM (in Table 1) that applies RL to abstractions based on programming or formal languages, AbstRaL - w/o GranulAR and AbstRaL - w/o Contexts (in Table 2) that conduct RL on abstractions with socratic CoT and de-contextualized formats, respectively. Our results indicate that **RL is sensitive to the change of abstract representations** when used to improve the reasoning robustness with abstraction.
> > >
> > > \
> > > We also thank the reviewer for pointing out more related work on the track of abstraction, and will cite these work in the camera-ready revision.

---

### Author Response · Authors · 2025-12-01
**Summary of Author Response and Author-Reviewer Discussion**

We thank the reviewers again for their valuable reviews and the AC in advance for their upcoming meta-review. Below we summarize the main concerns raised by each reviewer and how our initial responses and follow-up discussions address them.

\
**Reviewer AUyf** mainly concerns how our AbstRaL framework is novel regarding existing abstraction methods for math word problem solving. Importantly, we stress that our primary objective is to improve LLMs’ reasoning **robustness to distribution shifts, beyond just in-distribution** improvements. In that regard, we have verified in our ablation study that robust reasoning via abstraction has not been reliably learned by LLMs; see Section 5.1 and Table 2. Therefore, AbstRaL **bridges abstraction and robustness** in LLM reasoning, and contributes to an **improved schema** of abstraction on top of the existing works, to achieve a reliable learning of robust reasoning in LLMs.

Reviewer AUyf also made an inaccurate statement about the sufficiency of our experimentations, **stating incorrectly that only 2 LLMs were tested** when we tested in reality a diverse set of **9 LLMs** and **8 types** of distribution shifts.

Regarding the reviewer’s concern about our GranulAR data construction quality, we have clarified that our data construction does **not rely on** the large oracle LLM to distill reasoning traces, but is instead based on **re-writing of human-verified** samples, which naturally ensures the quality of decomposed reasoning steps.

After our initial response, reviewer AUyf corrected their review and updated their **rating from 2 to 4**, as indicated [here](https://openreview.net/forum?id=49vo7D9LbI&noteId=2AtWFbGB4s), and we believe our follow-up response after that has resolved their further concerns.

\
**Reviewer wvec**, similar to reviewer AUyf, raises the concern of whether our method relies on distilling superior reasoning capabilities from the oracle LLM. We have verified that our improvements are **not due to distilling** more superior reasoning knowledge from the oracle, and using a **weaker** LLM (e.g., Llama-3.1-8B-Instruct) as the oracle to construct the re-written data achieves **roughly the same** evaluation scores. We will add this in the final version.

Reviewer wvec also mis-judges that our experimental comparisons are primarily between AbstRaL (SFT+RL) and baselines that are either prompting-based (CoT-8S) or SFT-only (CoA, AoT). We clarify that AbstRaL **also outperforms** the RL baseline CoT-RL and the SFT+RL baseline SyReLM, and our ablated AbstRaL - w/o RL **still outperforms** SFT-only baselines CoA and AoT.

Reviewer wvec also concerns that the improvements of AbstRaL on OOD general reasoning benchmarks are marginal, and trade off the original GSM8K performance on some strong seed LLMs. We clarify that AbstRaL achieves **consistent improvements on all** our tested OOD datasets, with both marginal and more significant improvements (e.g., ~20% improvement on the broad-covered reasoning benchmark BBH when seeded with Qwen2.5-Math-7B-Instruct). We also view the **trade off as a strength** of our method, which mitigates LLMs’ overfitting to the existing in-distribution testing samples.

\
**Reviewers vEZ7** and wvec both asked about how our AbstRaL method would be applied to other math datasets beyond GSM and other types of reasoning scenarios. We clarify that our paper mainly aims to improve GSM robustness, as is the **most prominent** domain of focus in the current literature for robustness deficiencies in LLM reasoning. However, in our responses to reviewers, we still include an **example perspective** of how AbstRaL could be extended to other reasoning scenarios.

Reviewers vEZ7 also concerns how our AbstRaL is sensitive to noise added to the reasoning problem formulation. We clarify that as verified on the Distract perturbation in GSM-Plus dataset, AbstRaL achieves **superior robustness** to the interference of distractive noise.

Reviewer vEZ7, as indicated in their last follow-up comment [here](https://openreview.net/forum?id=49vo7D9LbI&noteId=Oqc3cOePBt), states that our response resolved or explained the points they raised, and the **overall rating of 8 is fair** for the contribution of our work.

\
**Reviewer XBfh** mainly concerns whether a simpler reasoning schema than GranulAR would achieve similar performance, and how our proposed symbolic distance reward can benefit the abstraction learning. We clarify that the schema of abstraction **significantly affects the resulting robustness** of learned abstract reasoning, and simpler schemas used in our baselines or ablations all fail to achieve comparable performance. In terms of the symbolic distance reward, our qualitative analysis in the response illustrates that the reward can **avoid learning erroneous reasoning shortcuts** coincidentally getting the correct answer, which thus benefits RL rollouts.

---

### Meta-Review · Area_Chair_YZNo · 2026-01-05

**Summary:**

This paper proposes AbstRaL, a novel framework designed to enhance the robustness of LLMs in mathematical reasoning tasks (specifically GSM). The core idea is to train the model to "abstract" reasoning problems—replacing specific values with placeholders—before solving them. Unlike previous works that rely solely on Supervised Fine-Tuning, this work demonstrates that learning this abstraction process via RL significantly improves performance and robustness against distribution shifts and distracting information.

During the review process, the reviewers generally appreciated the novelty of integrating RL with the abstraction process and the solid performance improvements. However, several concerns were raised, primarily regarding the sufficiency of the comparative baselines, the method's potential for generalization beyond the specific datasets tested, and clarifications on methodological details such as the reward design. Specifically, reviewers questioned whether the comparison against existing abstract reasoning methods was comprehensive enough to justify the claims of superior performance.

Although the discussion phase was unexpectedly interrupted, preventing a formal final consensus from all reviewers, I have thoroughly re-examined the paper, the reviews, and the authors' rebuttal to make a final decision. Regarding the primary concern about other abstract reasoning baselines, the authors clarified that they compared their method against CoA and AoT as representatives of SFT-only approaches, and SyReLM as a representative of SFT+RL approaches. I confirm that these baselines represent the current State-of-the-Art in the domain of abstract and symbolic reasoning, making the comparison both fair and sufficient. Furthermore, the rebuttal provided detailed explanations and additional context regarding the model's generalizability and training efficiency, which I believe effectively addresses the reviewers' doubts about the broader applicability of the method.

I am confident that had the discussion phase continued without interruption, the comprehensive clarifications provided by the authors would have fully resolved the reviewers' concerns. The paper addresses a critical problem in robust reasoning with a well-motivated method and rigorous execution. Therefore, I recommend accepting this paper for presentation.

**Reviewer Concerns:**

It appears that Reviewer AUyf still has concerns regarding novelty despite the authors' response. However, I believe that the authors' additional clarifications address the novelty issue to a reasonable extent.

Regarding the distillation issue raised by Reviewer wvec, the concern might not have been fully resolved, as the authors only mentioned conducting additional experiments without providing concrete quantitative results. However, the concerns regarding other baselines seem to have been adequately addressed.

**Reviewer Scores:**

Reviewer AUyf stated that they raised their score to 4 after a misunderstanding regarding the number of LLMs used was clarified. Had the concern regarding novelty been fully resolved, they likely would have raised their score to 6. Regarding Reviewer wvec, I believe they might have raised their score to 4 or 6, as their concerns appear to have been resolved.

---

### Decision · Program_Chairs · 2026-01-26

Accept (Poster)